# Autophagy facilitates adaptation of budding yeast to respiratory growth by recycling serine for one-carbon metabolism

Alexander I. May[1,2], Mark Prescott[3] & Yoshinori Ohsumi[1✉]

The mechanism and function of autophagy as a highly-conserved bulk degradation pathway are well studied, but the physiological role of autophagy remains poorly understood. We show that autophagy is involved in the adaptation of *Saccharomyces cerevisiae* to respiratory growth through its recycling of serine. On respiratory media, growth onset, mitochondrial initiator tRNA modification and mitochondrial protein expression are delayed in autophagy defective cells, suggesting that mitochondrial one-carbon metabolism is perturbed in these cells. The supplementation of serine, which is a key one-carbon metabolite, is able to restore mitochondrial protein expression and alleviate delayed respiratory growth. These results indicate that autophagy-derived serine feeds into mitochondrial one-carbon metabolism, supporting the initiation of mitochondrial protein synthesis and allowing rapid adaptation to respiratory growth.

[1] Cell Biology Center, Institute of Innovative Research, Tokyo Institute of Technology, 4259-S2-12 Nagatsuta-cho, Midori-ku, Yokohama 226-8503, Japan. [2] Tokyo Tech World Research Hub Initiative (WRHI), Institute of Innovative Research, Tokyo Institute of Technology, Yokohama, Japan. [3] Monash Biomedicine Discovery Institute, Department of Biochemistry and Molecular Biology, Monash University, Clayton campus, VIC 3800, Australia. ✉email: yohsumi@iri.titech.ac.jp

Autophagy is a bulk degradation mechanism targeting intracellular components, such as proteins, that occurs in almost all eukaryotes. Upon nutrient stress, autophagy is induced, initiating the delivery of cellular components to the lysosome (mammalian cells) or vacuole (yeast and plants), where they are degraded to basic metabolites and returned to the cytoplasm for reuse[1]. The isolation of cargo can occur apparently randomly (bulk autophagy) or in a selective manner (selective autophagy), the former requiring core Atg proteins and the latter additionally employing cargo-specific receptor proteins and the adaptor protein Atg11 to mark targeted cargo. These two arms of autophagy cooperate to ensure the efficient deployment of cellular resources under diverse conditions.

Notwithstanding its near-universal conservation throughout eukaryotes, the physiological function of autophagy remains poorly understood. The growth and survival of autophagy mutant cells are normal under standard growth conditions, although previous studies have shown that autophagy is essential for the maintenance of metabolite concentrations during periods of nutrient stress[2,3], is involved in development[4,5], and is also involved in cell cycle progression and survival[6–9]. However, how autophagy-derived metabolites are used by cells and the mechanisms underlying the physiological roles of autophagy remain unaddressed in the literature. To explore these questions, we assessed the growth of autophagy mutant yeast cells transitioning from glycolytic to respiratory growth, a condition requiring remodelling of much of the cell's metabolic machinery.

## Results

**Autophagy disruption results in delayed growth on respiratory media.** We set out to determine the effect of autophagy disruption on growth as an indicator of the physiological state of the cell. To this end, we assessed the growth of prototrophic strains of yeast on synthetic, minimal media. In all experiments, cells were precultured on synthetic glucose media supplemented with casamino acids, a casein hydrolysate providing a mixture of amino acids, prior to inoculation to experimental growth conditions (Supplementary Fig. 1a). We first assessed the glucose growth of wild-type cells and an $atg2\Delta$ mutant strain, the latter of which is defective for all identified forms of macroautophagy (Fig. 1a, left panel). Yeast cells grown on glucose are characterised by repressed expression of genes required for mitochondrial respiration, instead meeting energetic and biomass requirements through glycolysis[10,11]. Little difference was observed between wild-type and $atg2\Delta$ mutant cells throughout the initial logarithmic phase of growth. However, a defect in $atg2\Delta$ growth was observed during the post-log phase of growth, where the depletion of glucose triggers the expression of mitochondrial proteins and a concomitant increase in mitochondrial respiratory activity as cells shift to respiratory utilisation of ethanol[12]. We therefore assessed the growth of strains on ethanol media (Fig. 1a, right panel). In contrast to wild-type cells, the growth of $atg2\Delta$ cells is characterised by a prolonged delay in the onset of growth (lag phase). Growth of a strain deleted for $ATG15$, which encodes a protein required for the rupture of autophagic bodies following their delivery to the vacuole, reproduced the prolonged lag phase of $atg2\Delta$ cells, confirming the link between autophagic degradation and respiratory growth onset (Fig. 1b). To determine whether the prolonged lag phase of $atg2\Delta$ cells is specific to ethanol, we assessed the growth of wild-type and $atg2\Delta$ cells on a range of fermentable and non-fermentable carbon sources (Fig. 1c). Of those assessed, we observed the delay in autophagy mutant growth on all respiratory media but not on fermentable media, confirming that the onset of respiratory growth is perturbed in $atg2\Delta$ cells. We determined that the defective growth observed

was not due to key physiological differences (viability, petite frequency and cell size) between wild-type and $atg2\Delta$ cells prior to inoculation to growth media (Supplementary Fig. 1b–d), and verified the phenotype using an alternative prototrophic background strain of completely distinct lineage, D273-10B (Supplementary Fig. 2a–c). In addition, we confirmed that the delayed onset of growth is due to the fermentative to respiratory transition and not the absence of casamino acids from culture media (Supplementary Fig. 3). We also observed a delay in the growth-associated formation of small, newly-emergent cells in $atg2\Delta$ cultures grown on ethanol media (Supplementary Fig. 4), providing a further indication that disruption of autophagy delays the onset of respiratory growth.

**Non-selective autophagy is required for timely adaptation to respiratory growth.** As a core autophagy gene, the deletion of $ATG2$ blocks all known forms of bulk and selective autophagy. To broadly investigate what type of autophagy is required for adaptation to respiratory growth, we also assessed the growth of $atg1\Delta$ (lacking an alternative core autophagy gene), $atg11\Delta$ (lacking the selective autophagy adapter protein) and $atg32\Delta$ (lacking the mitophagy receptor protein) cells on ethanol media (Fig. 2a). The deletion of $ATG1$ closely reproduced the $atg2\Delta$ delay in onset of growth, whereas the disruption of $ATG11$ or $ATG32$ had no apparent effect on respiratory growth, suggesting that bulk, not selective, autophagy is involved in the onset of respiratory growth. Growth can be quantified in terms of the duration of the lag (adaptive) phase of growth, hereby referred to as $t_{lag}$, and the log-phase growth rate, $\mu_{log}$ (Fig. 2b). Compared to wild-type cells, there was only a very marginal increase in the $t_{lag}$ of $atg2\Delta$ cells grown on glucose, and growth rates were comparable between strains. On ethanol a prolonged $t_{lag}$ of was observed for $atg2\Delta$ but not $atg11\Delta$ cells that is statistically significant (Fig. 2c). We found no difference in $\mu_{log}$ between wild-type, $atg2\Delta$ and $atg11\Delta$ cells on ethanol (Fig. 2d). Further testing of strains lacking a representative selection of $ATG$ genes, including those for the various forms of selective autophagy (Fig. 2e, f and Supplementary Fig. 5), revealed that all grew comparably on glucose, but the delayed growth of $atg2\Delta$ cells was only reproduced among bulk autophagy mutant strains. In contrast, the deletion of non-core autophagy proteins resulted growth phenotypes similar to wild-type cells. These data therefore indicate that bulk but not selective autophagy is required for normal adaptation to respiratory growth.

**Transition to respiratory metabolism induces autophagy.** The respiratory growth adaptation defect in the absence of bulk autophagy suggests that exposure of cells to respiratory media is sufficient to induce bulk autophagy. To test this hypothesis, we assessed several measures of autophagic induction and delivery of cargo to the vacuole. The Cvt pathway is a constitutive selective autophagy pathway responsible for the delivery of precursor forms of vacuolar hydrolases synthesized in the cytoplasm, including Ape1[13]. The proform of this protein, prApe1, is cleaved to a mature form (mApe1) upon delivery to the vacuole[14], and in the absence of Atg11 this becomes solely dependent upon the induction of bulk autophagy[15]. The GFP-Atg8 cleavage assay, which assesses the autophagy-dependent cleavage of GFP-Atg8 within the vacuole[16,17], can also be used to monitor bulk autophagy progression in $atg11\Delta$ cells. During the early stages of growth on glucose, bulk autophagy activity was not observed (Fig. 3a). In contrast, inoculation to ethanol media resulted in the near-immediate onset of GFP-Atg8 cleavage, which was also observed in $atg11\Delta$ cells. Ape1 maturation was induced within 1.5 h following inoculation to ethanol media in $atg11\Delta$ cells,

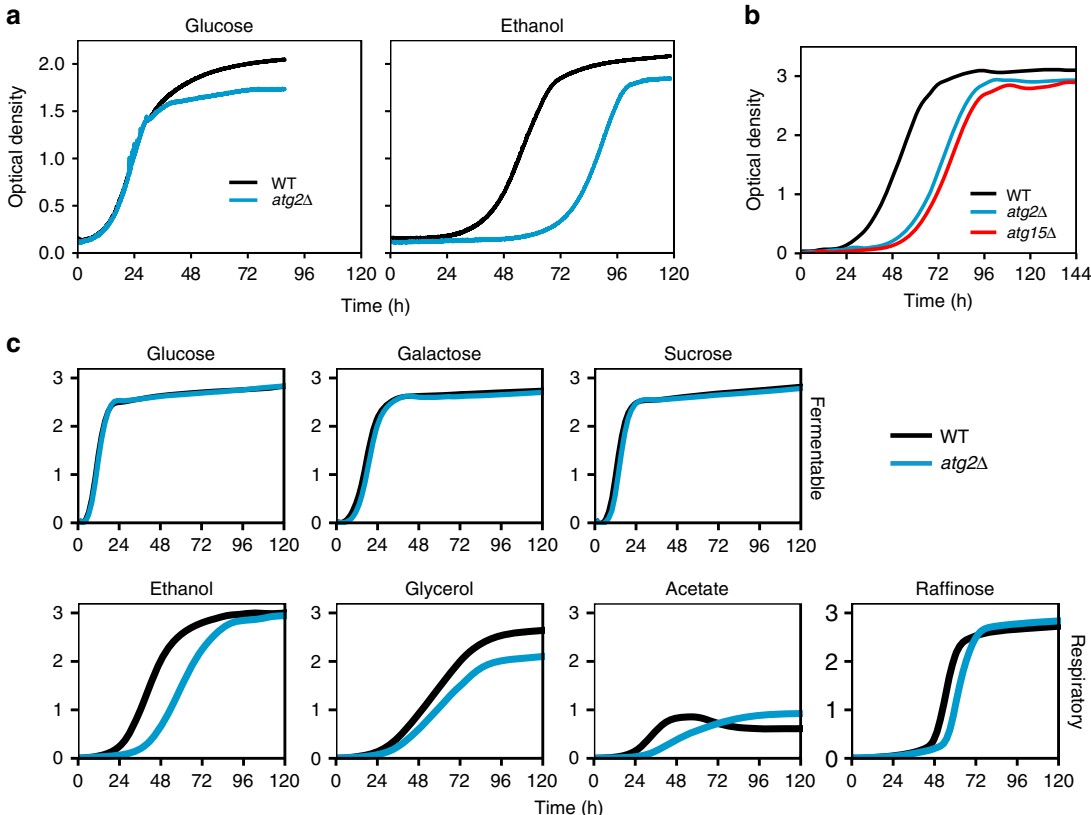

**Fig. 1 Autophagy mutant cells exhibit a delay in respiratory growth onset. a** Growth of wild-type (black) and autophagy mutant (blue) cells on synthetic media supplemented with glucose (left) or ethanol (right). **b** Growth of *atg15*Δ cells (red) on ethanol media. **c** Growth of WT (black) and *atg2*Δ (blue) cells on media supplemented with a range of carbon sources. Representative growth curves from at least $n = 3$ determinations are shown.

indicating the rapid induction of bulk autophagy. To confirm that these findings reflected autophagy induction, we also assessed formation of the pre-autophagosomal structure (PAS) by microscopy of cells expressing GFP-Atg8 (Fig. 3b). The PAS is known to accumulate in *atg2*Δ cells as Atg2 functions in the expansion of autophagosomal membranes downstream of PAS formation[18,19]. The formation of the PAS in *atg11*Δ cells at the earliest assessed time point in respiratory but not fermentative media (Supplementary Fig. 6a) provides further evidence that the induction of bulk autophagy occurs very rapidly upon exposure of cells to respiratory media.

In order to determine whether inoculation of cells to respiratory media alone is sufficient to induce autophagy, we also assessed the phosphorylation of the key regulator of autophagy induction, Atg13. The dephosphorylation of Atg13 results in autophagy induction by enhancing its interaction with Atg1, Atg17 and other downstream Atg proteins, leading to PAS formation[20]. Transient, partial dephosphorylation of Atg13 upon inoculation of cells to glucose media, which recovered within an hour of growth on glucose, was observed (Fig. 3c), likely as a result of the shift from casamino acids-containing preculture media to glucose media lacking casamino acids (Supplementary Fig. 1a). Indeed, cells shifted from a preculture condition lacking casamino acids showed persistent phosphorylation of Atg13 in glucose media, in contrast to the extensive dephosphorylation observed in respiratory media (Supplementary Fig. 6b). In contrast, inoculation of cells to ethanol media resulted in prompt, complete dephosphorylation that was not recovered, indicating that this condition strongly induces autophagy (Fig. 3c). We also observed marked dephosphorylation of Atg13 following inoculation to rich ethanol media containing yeast extract and peptone (Fig. 3d). These results suggest that the transfer of cells to

respiratory conditions is sufficient to strongly induce autophagy. In support of this conclusion, we found that shifting cells to media comprising carbon sources utilised by respiration resulted in robust dephosphorylation of Atg13 (Fig. 3e). Collectively, these data indicate that the exposure of cells to conditions requiring adaptation to respiratory growth is sufficient to induce autophagy.

**Supplementation of serine alleviates *atg2*Δ growth defect on respiratory media.** Catabolism and anabolism are linked by autophagy through the liberation of metabolites by autophagic degradation and their use by diverse cellular processes during stress. We next considered whether the prolonged lag phase observed in autophagy mutant cells might be due to a shortfall in autophagy-derived metabolite(s) required for adaptation to respiratory growth. This possibility was tested by supplementing a range of nutrients to ethanol media and assessing growth (Fig. 4a). The presence of 0.1% yeast extract, a complex media supplement, was sufficient to nearly completely recover the delay in *atg2*Δ growth onset, suggesting that a requirement for a yeast extract component(s) is not met in the absence of autophagy. Supplementation of a mixture of amino acids, which are present in significant quantities in yeast extract, was also able to nearly completely alleviate the prolonged lag phase of *atg2*Δ cells, strongly suggesting that amino acids are required by autophagy mutant cells during adaptation to respiration. Subsequent experiments revealed that the supplementation of serine to media resulted in the significant recovery of *atg2*Δ growth (Fig. 4b), and that concentrations of serine as low as 62.5 µM were able to effectively recover the growth of autophagy mutant cells (Fig. 4c and Supplementary Fig. 7a). For this reason, all amino acids are

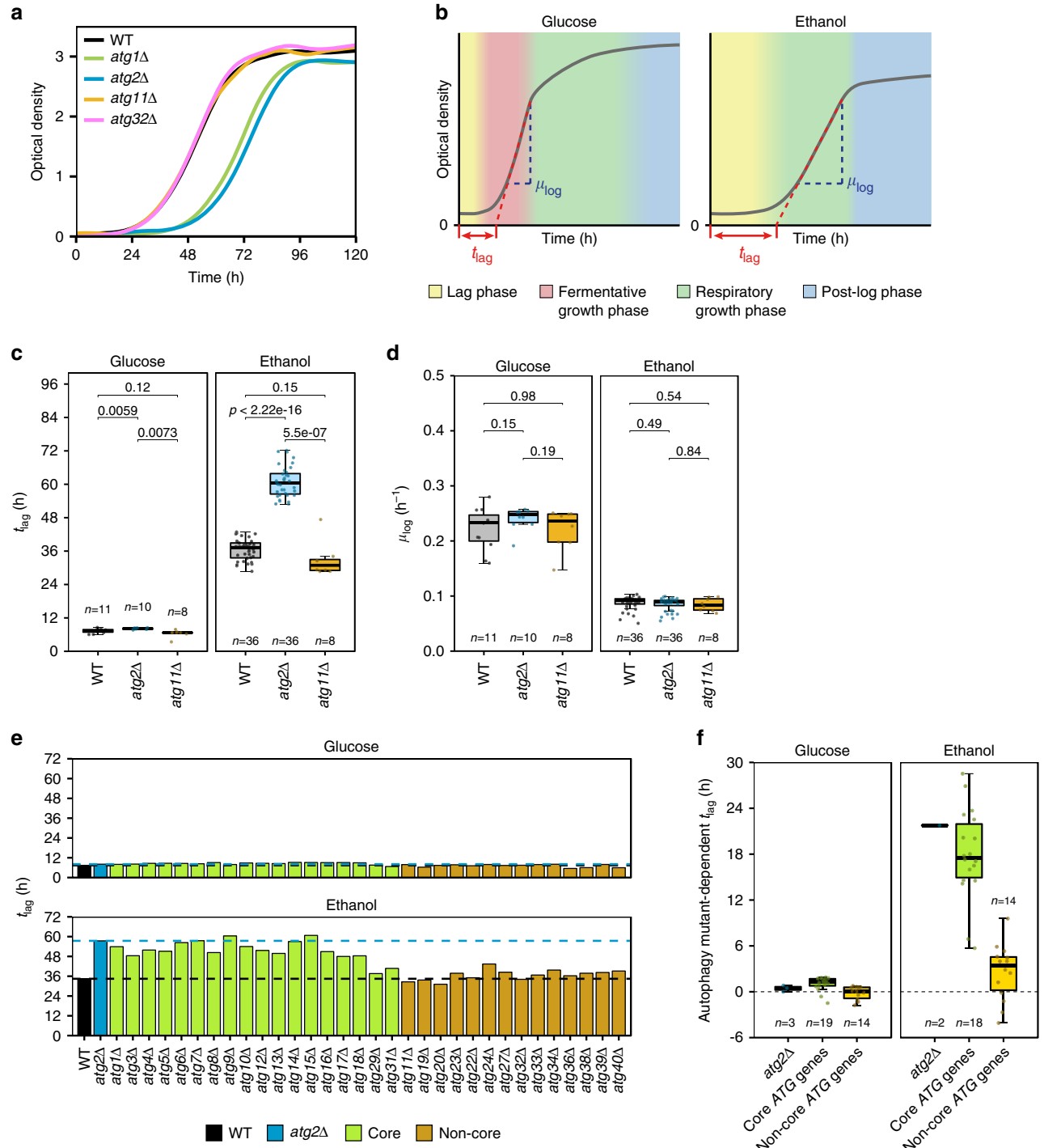

**Fig. 2 The delay in respiratory growth onset is attributed to bulk, not selective, autophagy. a** Growth of WT (black), *atg1*Δ (green), *atg2*Δ (blue), *atg11*Δ (yellow) and *atg32*Δ (mauve) cells on ethanol media. Data from a single experiment. **b** Schematic of phases of fermentative and respiratory growth outlining approach used to quantify parameters of growth in this study. Data are single representative WT growth curves for cells cultured on synthetic media in the presence of indicated carbon sources. **c** Statistical analysis of $t_{lag}$ and **d** $\mu_{log}$ for wild-type and *atg2*Δ cells. **e** Mutant-dependent $t_{lag}$ for a representative selection of *ATG* deletion mutant strains. Bars indicate the $t_{lag}$ of core autophagy (green) and non-core (yellow) autophagy deletion mutant strains. WT and *atg2*Δ $t_{lag}$ are indicated by broken lines. Data are from single determinations of growth. **f** Boxplot of mutant-dependent $t_{lag}$ for each autophagy deletion strain shown in **e** classed by their involvement in bulk or selective autophagy. Boxplots are shown as median (middle bar), 25th and 75th percentiles (upper and lower limits of the box) and 1.5 × interquartile range (whiskers), with sample sizes indicated on plots. Indicated *p*-values were calculated using the two-sided Student's *t*-test with Welch modification.

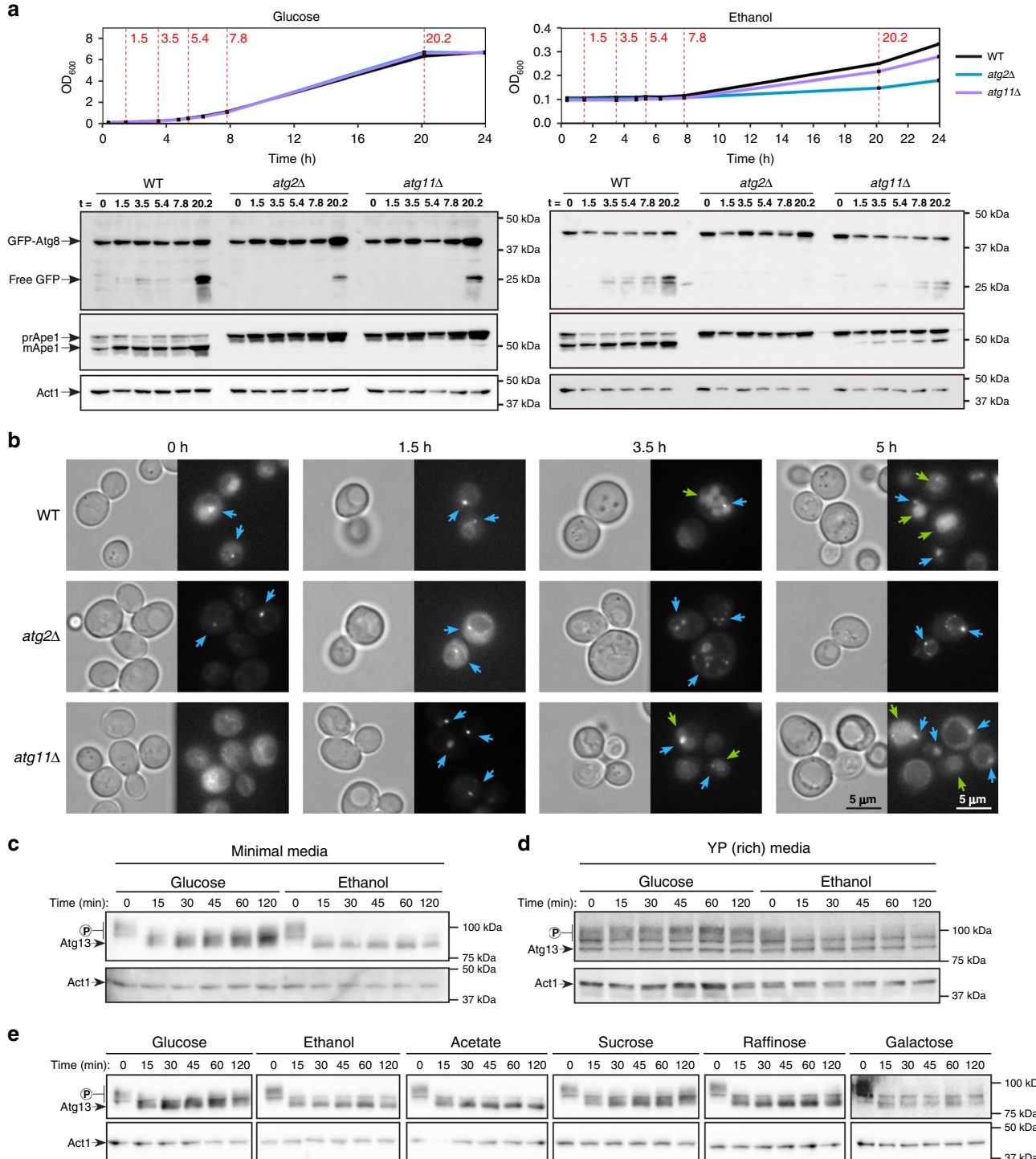

**Fig. 3 The presence of respiratory sources of carbon is sufficient to induce bulk autophagy. a** WT (black), *atg2Δ* (blue) and *atg11Δ* (purple) cells were sampled at indicated time points (upper panels) and GFP-Atg8 cleavage and Ape1 maturation assessed (lower panels) following inoculation to glucose or ethanol media. Shown are individual growth curves of cultures sampled for western blotting and microscopy in **b**. **b** PAS formation in WT, *atg2Δ* and *atg11Δ* cells expressing GFP-Atg8 inoculated to ethanol media. Blue arrows indicate PAS formation, whereas green arrows show accumulation of diffuse GFP signal indicating delivery of cargo to the vacuole by autophagy. **c** Phosphorylation of Atg13 in wild-type cells grown on glucose or ethanol media. **d** Phosphorylation of Atg13 in wild-type cells grown on yeast extract-peptone (YP) rich media containing glucose or ethanol as carbon sources. **e** Phosphorylation of Atg13 on synthetic media containing a range of carbon sources. Data are from single representative experiments that were reproduced twice (**b**, **d**, **e**) or three times (**a**, **c**).

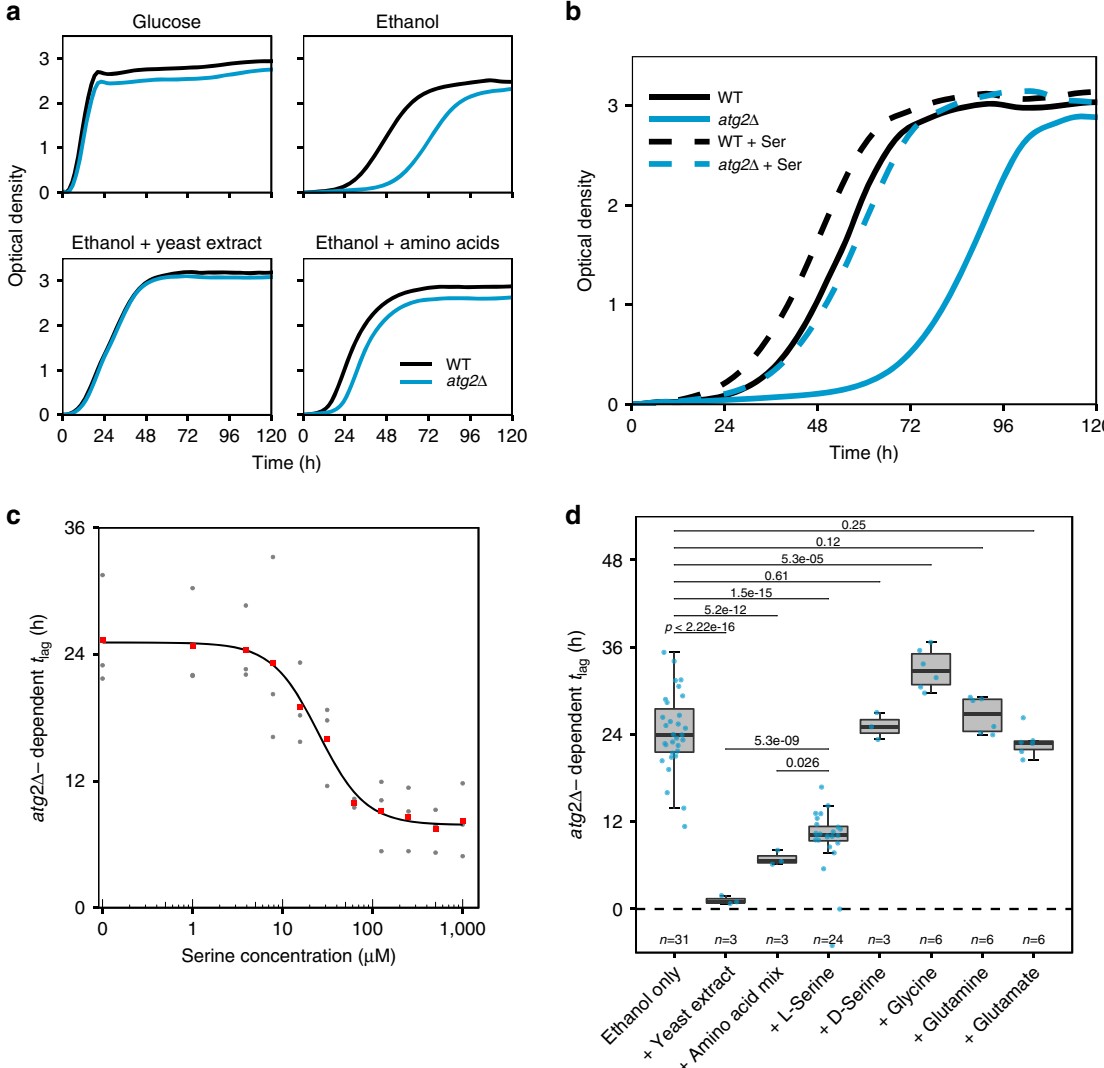

**Fig. 4 Serine supplementation alleviates _atg2Δ_-dependent _t_lag on ethanol media. a** Wild-type (black) and _atg2Δ_ (blue) cells were grown on glucose media (upper left) or ethanol media with indicated supplements. **b** Growth of WT (black) and _atg2Δ_ (blue) cells on ethanol media with (broken lines) or without (solid lines) supplemented serine. **c** Dose–response curve of serine concentration vs _atg2Δ_-dependent _t_lag. Mean values (red points) of three independent determinations (grey points) is shown, with logistic fitting to mean values. **d** Recovery of autophagy-dependent _t_lag in response to the provision of ethanol media with indicated supplements. Representative individual growth curves are shown in **a** and **b**. Boxplots are shown as median (middle bar), 25th and 75th percentiles (upper and lower limits of the box) and 1.5 × interquartile range (whiskers), with sample sizes indicated on the plot. Indicated p-values were calculated using the two-sided Student's t-test with Welch modification.

supplemented herein at 62.5 μM. Growth recovery was not observed for the D-enantiomer of serine, and the supplementation of glycine, which is metabolically linked to serine through similar synthetic and catabolic mechanisms, resulted rather in a marginal exacerbation of _atg2Δ_ _t_lag (Fig. 4d). These effects do not appear to be linked to bulk nitrogen catabolism as the supplementation of glutamine and glutamate, which are central metabolites in nitrogen metabolism, had little effect on growth. We excluded the possibility that serine exerts an effect of autophagy induction, finding that the supplementation of serine to culture media had no effect on autophagy induction (Supplementary Fig. 7b). Further, iron supplementation, which was previously identified as critical for post-diauxic growth in autophagy mutant cells[21], had no effect on the timing of _atg2Δ_ growth onset (Supplementary Fig. 8). Together, these data suggest that in the absence of autophagy, cells experience an acute shortfall in intracellular serine, which perturbs the cell's ability to adapt to respiratory growth, delaying the onset of _atg2Δ_ growth.

**Mitochondrial respiration is perturbed in the absence of bulk autophagy.** The characteristic delay in the onset of autophagy mutant growth suggests the perturbation of mitochondrial respiration in the absence of autophagy. The expression of mitochondrial proteins, including electron transport chain components, is a key feature of adaptation to respiratory growth, the progress of which is reflected in increased oxygen consumption. In contrast to the rapid increase in oxygen consumption observed in wild-type cells, mutant cell respiration remained low during the prolonged _t_lag following inoculation to ethanol media (Fig. 5a). This result was reproduced in the presence of FCCP, an uncoupler that gives an indication of the maximum respiratory capacity of mitochondria. The supplementation of serine was able to rescue the delay in autophagy mutant respiration: similar increases in oxygen consumption were observed in both strains. We also assessed mitochondrial membrane potential ($\Delta\Psi_m$) in cells using $DiOC_6$, a lipophilic dye that accumulates in mitochondria with high $\Delta\Psi_m$. $\Delta\Psi_m$ is elevated in functional

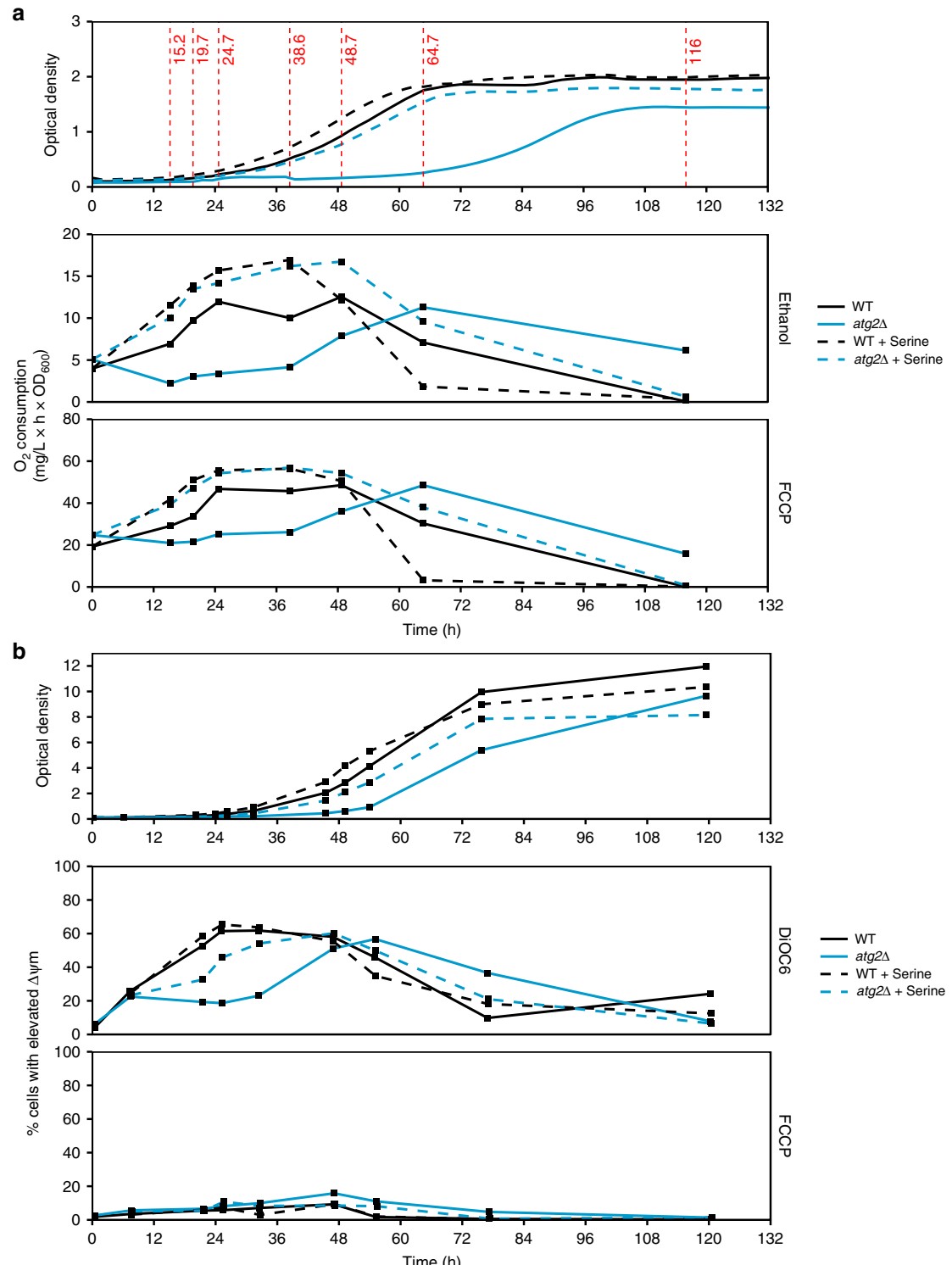

**Fig. 5 Mitochondrial respiration is perturbed in the absence of bulk autophagy. a** Oxygen consumption rates of WT (black) and *atg2Δ* (blue) cells at various points of respiratory growth. The growth of strains is shown in the top panel, whereas substrate (ethanol-dependent) oxygen consumption and decoupled (+FCCP) oxygen consumption rates are shown in the lower panels. Cells grown on serine-supplemented media are indicated by dashed lines. **b** The proportion of of WT and *atg2Δ* cells characterised by high mitochondrial membrane potential ($\Delta\Psi_m$), with strains and conditions as indicated in **a**. Membrane potential was determined by flow cytometry of DiOC6 stained cells, as described in the methods. The gating strategy is shown in Supplementary Fig. 12. All data are single determinations ($n = 1$).

mitochondria, and acquisition of $\Delta\Psi_m$ is a hallmark of adaptation to respiratory growth conditions. We found that a subpopulation of newly emerged cells characterised by high $\Delta\Psi_m$ appeared in wild-type but not *atg2Δ* cells soon after inoculation to ethanol media, whereas *atg2Δ* cells experienced this shift later (Fig. 5b). The acquisition of $\Delta\Psi_m$ correlates well with both the onset of wild-type cell growth and the acquisition of respiratory capacity, providing further evidence of a fundamental delay in the

preparation of mitochondrial metabolism for respiratory growth. As for oxygen consumption, the supplementation of serine resulted in the nearly full recovery of $\Delta\Psi_m$ in $atg2\Delta$ cells to wild-type levels under the same conditions.

**The autophagy growth defect is linked to mitochondrial one-carbon metabolism.** Serine feeds into a range of cellular pathways, including the reactions of one-carbon metabolism. Serine is the predominant donor to this pathway, providing one-carbon units derived from its carboxyl group through a condensation reaction (Supplementary Fig. 9a)[22]. This one-carbon unit is accepted and carried by tetrahydrofolate (THF) through duplicated cytoplasmic and mitochondrial arms of one-carbon metabolism that are linked, with one-carbon units thought to flow from mitochondria to the cytoplasm (Fig. 6a)[23].

While the majority of mitochondrial proteins induced upon glucose depletion are encoded in nuclear DNA, a number (in yeast, eight) are encoded by the mitochondrial genome[24]. The initiation of mitochondrial translation is mediated by mitochondrial initiation factor 2 (mIF2), which plays a critical role in bringing mitochondrial ribosomes, mRNA and initiator tRNA together, thereby facilitating translation. The formylation of initiator tRNA (tRNA$^{fMet}$) in the mitochondrion is dependent upon mitochondrial one-carbon metabolism[25,26]. Fmt1 catalyses the attachment of the one-carbon-derived formyl group to tRNA$^{fMet}$, which enhances its interaction with mIF2, thereby promoting mitochondrial translation in a manner reminiscent of prokaryotic translation initiation. However, other proteins have also been shown to facilitate the interaction between mIF2 and the translational machinery, allowing Fmt1-independent translation. These proteins include the ribosomal protein Rsm28[27,28], although exclusive Rsm28-dependent translation is less efficient[29], and Msc6[30] and Aep3[31]. This redundancy results in the blockage of mitochondrial translation and respiratory growth only when the $RSM28$, $MSC6$ or $AEP3$ genes are deleted in $fmt1\Delta$ cells.

We next asked whether the delay in growth onset observed in autophagy mutant cells could be attributed to a perturbation in mitochondrial one-carbon metabolism. $fmt1\Delta$ cells were characterised by a near-$atg2\Delta$ growth phenotype, and the deletion of $MIS1$, which encodes the enzyme one step upstream of Fmt1, also resulted in an $atg2\Delta$-like delay in growth onset on respiratory media (Fig. 6b). In contrast to $atg2\Delta$ cells, however, the recovery of growth onset delay upon serine supplementation observed in $mis1\Delta$ and $fmt1\Delta$ cells was reduced, indicating that while the provision of this amino acid is able to recover the growth of autophagy mutant cells, this effect is markedly reduced in the absence of one-carbon metabolic enzymes. A further prolongation in $t_{lag}$ was observed in $mis1\Delta\ atg2\Delta$ and $fmt1\Delta\ atg2\Delta$ cells, although serine supplementation only recovered the growth of these strains to $mis1\Delta$ and $fmt1\Delta$ growth phenotypes, respectively. Meanwhile, the deletion of $SHM1$, which encodes the mitochondrial serine hydroxymethyltransferase, resulted in a slower respiratory growth phenotype but was not characterised by a delay in growth onset, and a $shm1\Delta\ atg2\Delta$ strain showed comparable growth onset to the $atg2\Delta$ strain (Supplementary Fig. 10). These data suggest that cytosolic serine hydroxymethyltransferase may be able to supply one-carbon units to the mitochondrion in the absence of Shm1.

As these results suggest that the serine-dependent recovery of $atg2\Delta$ growth is mediated by mitochondrial initiator tRNA modification, we next set out to directly assess initiator tRNA formylation by means of northern blotting with a radioisotope-labelled oligonucleotide specific for mitochondrial initiator tRNA (Supplementary Fig. 9b)[29]. In cells grown on ethanol, a strong band corresponding to formylated Met-tRNA (fMet-tRNA$^{fMet}$) appeared

within 24 h in wild-type cells (Fig. 6c). In contrast, $fmt1\Delta$ and $mis1\Delta$ cells showed a clear inability to formylate tRNA$^{fMet}$ (Supplementary Fig. 11a); this reflects the complete absence of initiator tRNA formylation in these strains. A clear lag in the formylation of initiator tRNA was observed in $atg2\Delta$ that closely matched the delayed growth of this strain. In the presence of serine, a marked acceleration in formylation was observed in $atg2\Delta$ cells at a rate comparable to wild-type cells. Formylation in $shm1\Delta$ cells was comparable at all assessed time points, with a slight reduction in fMet-tRNA$^{fMet}$ upon the commencement of growth (Supplementary Fig. 11b), a phenotype that closely reflects the growth of this strain and confirms the redundancy of serine hydroxymethyltransferase activity in cells. This result is consistent with a model whereby initiator tRNA formylation is still able to occur in $atg2\Delta$ cells, but is delayed due to the lack of precursor.

Glycine has previously been reported to act as another precursor for mitochondrial one-carbon metabolism, primarily through the activity of the mitochondrial glycine cleavage complex, which is able to supply $CH_2$-THF in this organelle[32]. As described above, glycine supplementation did not rescue the delayed onset of $atg2\Delta$ cells (Fig. 4d), instead resulting in a slight, non-specific inhibition of cellular growth. We tested whether glycine supplementation is able to feed into mitochondrial one-carbon metabolism by assessing initiator tRNA formylation (Supplementary Fig. 11c). No increase in initiator tRNA formylation was observed under the conditions employed in this study, providing a further indication that tRNA$^{fMet}$ formylation is corresponds with the onset of respiratory growth. The inhibitory effect of glycine on growth may be due to product inhibition of the mitochondrial and cytosolic serine hydroxymethyltransferases, thereby perturbing cellular one-carbon metabolism.

In order to confirm that the delay in formylation observed in $atg2\Delta$ cells results in delayed expression of mitochondrial proteins, we directly assessed mitochondrial protein translation in vivo using $^{35}$S-labelled amino acids (Fig. 6d). As expected[33], translation of mitochondrially-encoded proteins was low in both wild-type and $atg2\Delta$ cells during preculture on glucose. Wild-type cells showed robust induction of mitochondrial translation after 24 h of culture on ethanol media, whereas $atg2\Delta$ cells showed little synthesis of mitochondrial translation products. Consistent with increased initiator tRNA formylation, the supplementation of serine resulted in the clear restoration of $atg2\Delta$ mitochondrial translation. Further, mitochondrial translation in $atg2\Delta$ cells was recovered without supplemented serine to near wild-type levels after 48 h, at which point respiratory growth has commenced. Translation of mitochondrial proteins in $shm1\Delta$, $fmt1\Delta$ and $mis1\Delta$ strains (Supplementary Fig. 11d) was consistent with the growth and formylation phenotypes of these strains. $shm1\Delta$ cells were characterised by a near-WT level of protein expression, whereas $fmt1\Delta$ and $mis1\Delta$ cells showed notably lower rates of protein expression that were not recovered by serine supplementation. Considering this evidence, we propose a model (Fig. 7) whereby serine is a growth-limiting factor during adaptation to respiratory growth, and that its utilisation by mitochondrial one-carbon metabolism facilitates mitochondrial translation initiation. Mitochondrial translation increases the levels of key electron transport chain proteins, boosts respiratory capacity and facilitates respiratory growth. While autophagy is able to supply serine to overcome the shortfall encountered during the early stages of respiratory growth, supplemented serine is required to restore the growth of autophagy mutant cells.

## Discussion

Despite what appears to be the universal conservation of autophagy amongst eukaryotes, both the physiological role of

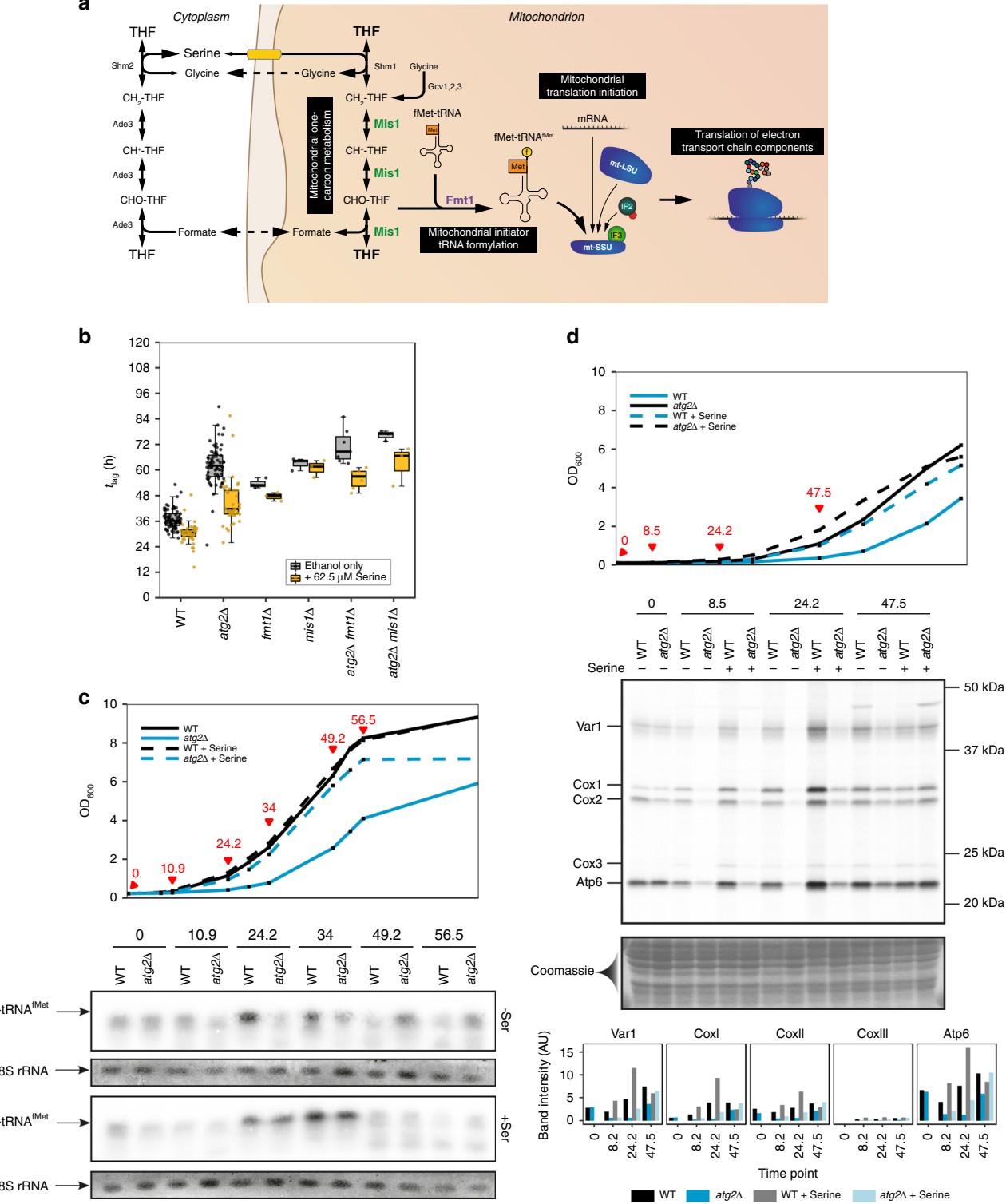

**Fig. 6 Autophagy-dependent $t_{lag}$ is caused by delayed mitochondrial translation initiation. a** Overview of one-carbon metabolism and the mechanism of protein translation initiation in mitochondria. **b** Boxplots of indicated strain $t_{lag}$ on ethanol media in the presence or absence of serine. Boxplots are shown as median (middle bar), 25th and 75th percentiles (upper and lower limits of the box) and 1.5 × interquartile range (whiskers), with sample sizes indicated on the plot. **c** Formylation of mitochondrial initiator Met-tRNA for WT and atg2Δ cells grown in the absence or presence of supplemented serine. Samples were collected at the time points indicated in the upper panel and subjected to 32P northern blotting of whole-cell RNA extracts. The formylated species (fMet-tRNA^fMet) is indicated. 18 S RNA is provided as a loading control **d** 35S labelling of mitochondrial translation products in WT and atg2Δ cells grown on ethanol media in the absence or presence of serine. Samples were collected at the time points indicated in the upper panel before labelling was conducted in the presence of the cytosolic translation inhibitor cycloheximide. Coomassie staining is provided as a loading control. Representative data from two independent experiments is shown.

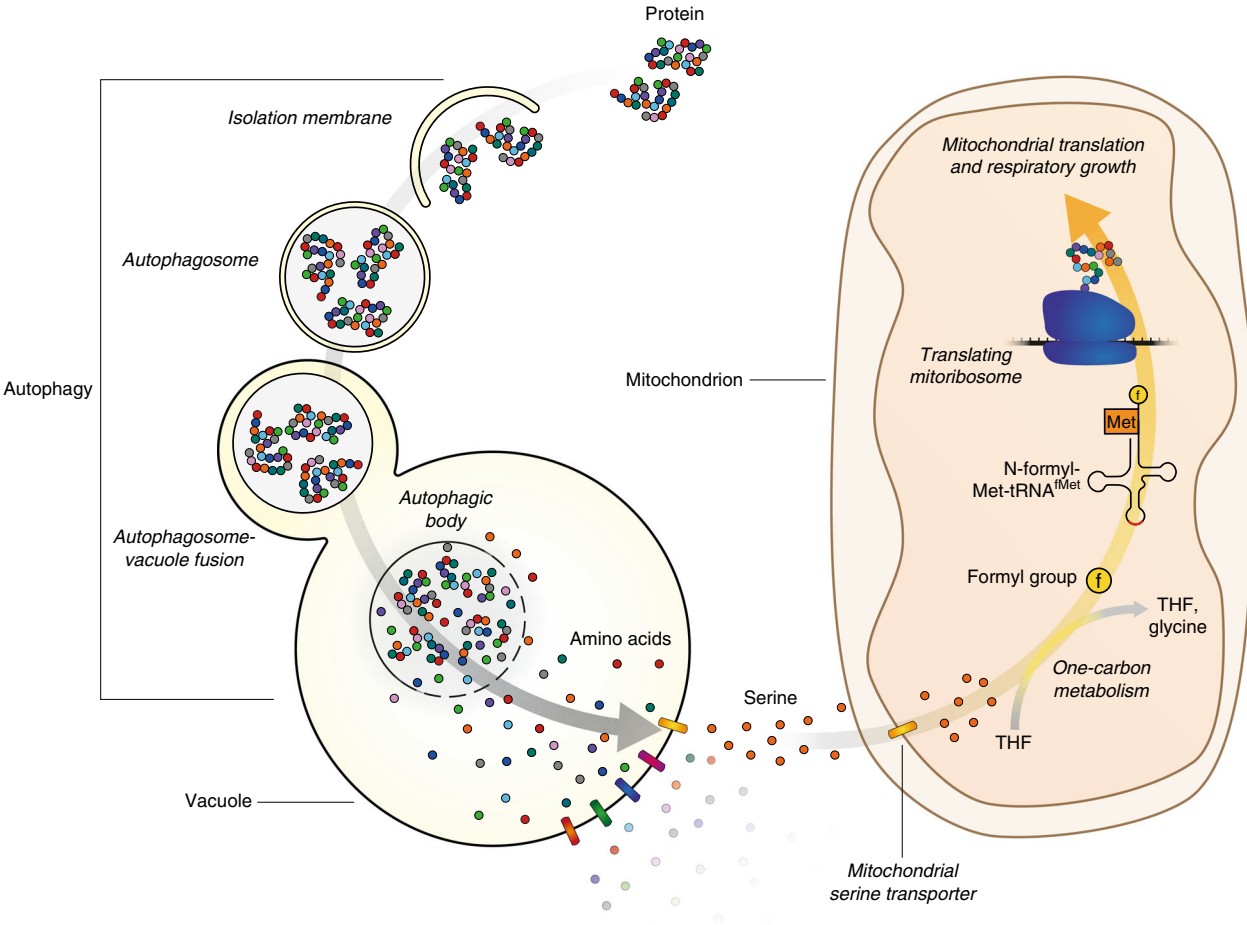

**Fig. 7 Serine is provided by bulk autophagy during adaptation to respiratory growth to support mitochondrial translation and function.** Upon inoculation to respiratory media, autophagy is induced, resulting in the degradation of proteins to amino acids in the vacuole. Amino acids are then returned to the cytoplasm by vacuolar effluxers (coloured bars). Following transport into mitochondria by the yet poorly characterised mitochondrial serine transporter, serine acts as a precursor for the reactions of mitochondrial one-carbon metabolism, yielding a formyl group that supports the formylation of initiator tRNA and translation initiation in this organelle. In the absence of bulk autophagy, the cytosolic pool of serine is depleted, limiting the flow of serine into mitochondria, which delays mitochondrial translation initiation and growth on respiratory media.

autophagy and the mechanisms that underpin these physiological functions remain poorly understood. In this report, we utilise prototrophic strains and a controlled glucose to ethanol transition to provide clear evidence that the shift from glycolytic to respiratory growth is addressed through the activation and progression of bulk autophagy, which stimulates mitochondrial translation by liberating serine. Our group has previously shown that autophagy is induced during the diauxic shift and that under these conditions autophagy mutant cells require iron to commence post-logarithmic growth[21]. Adachi et al. also found that respiratory capacity is essential for autophagy induction under carbon starvation conditions, most likely due to the provision of ATP in physiologically challenging conditions[34]. However, the conditions employed in these studies do not directly interrogate whether autophagy is induced in direct response to the transition to respiratory metabolism.

The link between autophagy, one-carbon metabolism and respiration has likely been overlooked due to the wide use of auxotrophic strains and rich culture media in autophagy research. As serine is produced from the bulk glycolytic intermediate glycerol-3-P during growth on glucose[35] and cellular demand for serine is high during respiration, the abrupt shift from glycolysis to respiration likely results in a critical and persistent shortfall in serine. This appears to be particularly important as serine is

implicated in mitochondrial one-carbon metabolism, where it serves as a precursor metabolite for the initiation of mitochondrial translation. Together, the findings reported in this study show that autophagy is a key source of serine during adaptation to respiratory growth, and further that formylation of mitochondrial initiator tRNA is required for optimal adaptation to respiratory growth conditions.

While previous studies have indicated that serine is rapidly depleted during nitrogen starvation and rapamycin treatment[36,37], it is unlikely that serine is completely depleted from cells under the conditions employed in this study, even in the absence of autophagy. Serine in the cytoplasm must be transported into mitochondria, probably by the recently identified mitochondrial serine transporter[38]. The affinity of this transporter, which has not yet been characterised in yeast, may determine a cytoplasmic serine concentration threshold under which serine does not enter mitochondria due to the low affinity nature of the transporter. The liberation of serine by autophagy may raise the cytosolic concentration of this amino acid beyond such a critical threshold, ensuring that the amino acid is transported into mitochondria in sufficient quantities to supply translation initiation. While the total intracellular concentration of metabolites is often considered in studies of cellular metabolism, this study therefore indicates that the localisation of

metabolites within cellular compartments may have important physiological implications for cells, especially during adaptation to changing environmental conditions.

As serine is consumed by numerous pathways, such as lipid synthesis, redox balancing and bulk protein synthesis, competition for limited serine exacerbates the limitation of supply during adaptation to respiration, even when not considering the import of serine to mitochondria. The relative demand of these pathways likely changes as cells adapt to the surrounding environment, and the ubiquitin-proteasome system and other biosynthetic pathways found in prototrophic strains may also generate serine for anabolic pathways. The non-specific recovery of growth in all strains upon serine supplementation observed in Fig. 6b can likely be attributed to the supply of this limiting amino acid to such alternative pathways. Together with the less efficient mitochondrial translation initiation mechanisms attributed to Rsm28, Aep3 and Msc6, these factors may explain why after the prolonged period of adaptation to ethanol media, autophagy mutant cells are apparently able to grow at a comparable rate to wild-type cells.

A previous study has indicated that mitochondrial translation is coupled to cytoplasmic translation through mitochondrial transcription factors that are encoded in the nuclear genome[39], raising the possibility that the defect in mitochondrial translation may reflect reduced cytoplasmic translation in the absence of autophagy. However, the recovery of $atg2\Delta$ growth is serine-specific: the supplementation of amino acids other than serine, including glutamine and glutamate (Fig. 4d), which are central players in amino acid metabolism, did not alleviate the delayed onset of autophagy mutant growth. Our data raise the prospect that the lack of respiration in autophagy mutant cells results in a physiological scenario (depletion of intracellular ATP and TCA cycle-derived metabolites) that indirectly hinders cytoplasmic translation without the need for a dedicated protein machinery. The possibility of such indirect regulation of cytoplasmic translation is supported by our observation that the growth rate of autophagy mutant cells is normal following the activation of oxidative phosphorylation and associated onset of growth (Fig. 2d).

This study also provides clear evidence that autophagy is induced upon exposure to respiratory growth conditions, even in the presence of 'rich' media supplements such as yeast extract and peptone. Atg13 dephosphorylation in particular indicated that a shift to respiratory media is a strong induction signal for autophagy, with persistent dephosphorylation occurring within 15 min of the shift to ethanol or other respiratory growth media. In the literature, the induction of autophagy is generally characterised as being regulated by TORC1[40] and amino acid availability[41,42]. However, as autophagy is induced even in the presence of replete amino acids (Fig. 3d), respiratory growth-induced autophagy may hint at intricate integration of upstream signals that control TORC1 activity, such as PP2A, Gtr-/Pib2-dependent and Snf1 signalling pathways[34,43–47]. As changes in carbon source availability are often encountered by yeast cells in the wild, such complex signalling appears likely.

This study has demonstrated how under a defined metabolic scenario (adaptation to respiratory growth), autophagy is important to overcome a shortfall in a critical metabolite (serine), thereby allowing cells to adapt to changing nutrient conditions. We broadly interpret these data as indicating that under various transitional and stress conditions, autophagy ensures that key adaptive pathways have access to sufficient precursors, thereby allowing the most efficient deployment of cellular resources in response to fluctuations in the surrounding environment. As the perturbation of both one-carbon metabolism[48] and autophagy[49] have been implicated in the progression of cancer, the demonstrated link between autophagy and one-carbon metabolism

examined in this study has implications for studies in mammalian systems. We anticipate that reported links between autophagy and various diseases will be understood in further detail as the intricate role of autophagy in metabolite homoeostasis becomes further clarified.

## Methods

**Experimental model.** The haploid parental strains X2180-1B and D273-10B, which are prototrophic strains of independent pedigree, were used. Mutants of these strains were derived by PCR-mediated homologous recombination whereby the entire locus of interest was replaced by a selection marker (for deleted genes) or supplemented by a tag with a selection marker (for tagged strains)[50,51]. Stable integration of the amplified PCR fragment was confirmed by PCR and by western blotting, where appropriate. Strains were otherwise handled according to established protocols[52]. Unique biological materials described in this study are available upon reasonable request.

Cell culture involved two stages of preculture on SDCA media (0.17% yeast nitrogen base without amino acids or ammonium sulphate (Difco), 0.5% casamino acids (Difco), 0.5% ammonium sulphate (Nacalai Tesque) and 2% glucose (Nacalai Tesque)) for 24 h each, followed by inoculation to subsequent media, which unless otherwise indicated was synthetic minimal media (0.17% yeast nitrogen base without amino acids or ammonium sulphate (Difco), 0.5% ammonium sulphate (Nacalai Tesque) and 2% carbon source), as detailed in Fig. S1A and with supplements as described in the text. Unless otherwise indicated, inoculations were at a culture density of $OD_{600} = 0.1$. Batch growth of experimental cultures was conducted at 30 °C and 180 rpm in baffled polycarbonate flasks. Supplements were made fresh in ultrapure water and filter-sterilised using a 0.22 μm PVDF membrane (Millipore) before addition to cultures. Automated growth was conducted using a small-scale rocking incubator or a 700 mL capacity fermenter, as described below.

For assessment of phosphorylation, cells were precultured as described above before inoculation to experimental conditions at $OD_{600} = 1$ to facilitate collection of cells for analysis.

In general, cells were collected by centrifugation for subsequent analyses before snap freezing in liquid nitrogen. Where pelleting of cells was difficult (in synthetic media at very low densities of cells), cells were collected by filtration on a 0.45 μm pore size mixed cellulose ester membrane (Advantec).

**Automated growth of cells.** Cells were inoculated to growth media as described above and grown in a 700 mL capacity fermenter (ABLE corporation) or a compact rocking incubator (Advantec). Fermenter-grown cells were cultured using a 700 mL capacity fermenter vessel mounted on a BMZ-P medium-scale fermenter (ABLE corporation, Tokyo) equipped with a filtered air pump and sparger, temperature regulator, a rotating shaft with fins for agitation and a pH sensor, all controlled and recorded electronically. Data from a Finesse 1 optical density probe was recorded using a TruCell2 transmitter connected to a computer (Finesse Corporation). Agitation was set at 440 rpm and 300 mL/min of filtered air was supplied to cultures maintained at 30 °C. 0.002% adecanol (Adeka) was added to fermenter cultures as an anti-foaming agent.

Compact rocking incubator-grown cells were cultured in a 5 mL capacity L-shaped tube within a TVS062CA incubator at 70 rpm and 30 °C. Culture optical densities were recorded automatically using Advantec TCS062CA communications software (Ver. 100103) during a 15 sec pause at 10 min intervals.

**Lysis of cells for polyacrylamide gel electrophoresis.** A urea-based extraction method, modified from a previous study[53], was used to prepare whole cell lysates for electrophoresis. Briefly, 1 $OD_{600}$ unit of cells was removed from culture, pelleted, immediately frozen on liquid nitrogen and stored at −80 °C. Cells were resuspended in ice-cold 10% trichloroacetic acid and left to incubate on ice for 10 min before pelleting by centrifugation (17,800 g, 4 °C, 15 min). Cell pellets were then resuspended in room temperature HU buffer solution (8 M Urea, 200 mM sodium phosphate buffer (pH6.8), 5% w/v sodium dodecyl sulphate (SDS), 100 mM ethylenediaminetetraacetic acid (EDTA), 15 mM Dithiothreitol (DTT) and traces of bromophenol blue sufficient for colour) and approximately half a volume of zirconium-coated beads (Yasui Kikai YZB05, diameter 0.5 mm) was added to the solution. Cells were then lysed by vigorous agitation in a FastPrep (MP Biomedical, Tokyo, Japan) for 30 seconds at room temperature. The liquid fraction of the lysate was isolated, incubated at 65 °C for 15 min, vortexed for 2 min and finally centrifuged at 17,800 g for 15 min at room temperature before the supernatant was subjected to electrophoresis.

**Polyacrylamide gel electrophoresis and western blotting.** Protein samples were separated by sodium dodecyl sulphate polyacrylaminde gel electrophoresis (SDS-PAGE) using a Bio-Craft BE-260 wide-format electrophoresis system (Bio-Craft KK, Tokyo, Japan). 12% gels were used for GFP-Atg8 cleavage assays, Ape1 maturation assays and mitochondrial protein samples, whereas phosphorylation was determined on 8% gels. Proteins separated by SDS-PAGE were transferred to PVDF membrane by a semi-dry method using a BioRad Trans-Blot Turbo

semi-dry transfer system, according to the manufacturer's instructions. Blocking and subsequent probing were performed using 1% skim milk in TBST solution.

**Microscopy**. Fluorescence microscopy images were acquired as multiplane z-stack images at 0.2 μm intervals over 6 μm on an inverted fluorescence microscope (Olympus IX81) equipped with an electron-multiplying CCD camera (ImageEM C9100-13) and 150× objective lens (UAPON 150× OTIRF, NA/1.45; Olympus). A 488 nm laser (Coherent) was used for excitation at 10 mW. Fluorescence was filtered with a Di01-R488/561-25 dichroic mirror and Em01-R488/568-25 bandpass filter (Semrock) and was separated from mCherry signal using a U-SIP splitter (Olympus) housing a DM565HQ dichroic mirror (Olympus). Fluorescence was further filtered with an FF02-525/50-25 bandpass filter (Semrock). Initial image processing was performed in Metamorph software (Molecular Devices) before individual planes of the z-stack were deconvolved and cast as a single maximum intensity image using AutoQuant X3 software (Ver. 3.0.5, Media Cybernetics) with provided optical parameters for the UAPON 150× OTIRF objective lens. Images were cropped for presentation using ImageJ software (Fiji Ver. 1.52p)[54].

**Assessment of oxygen consumption of cells**. To determine the oxygen consumption of cells, equivalent densities of cells (10 $OD_{600}$ units) were removed from culture and washed twice in a pH 5.8 potassium phosphate buffer. Cells were resuspended in the same phosphate buffer at 30 °C and introduced into a cuvette with continuous agitation. Oxygen concentration of the cell suspension was determined by a non-invasive oxygen sensor, sensor spots and a Fibox 3 transmitter (Precision Sensing GmbH). The cuvette was sealed throughout measurements and ethanol (final 2%) or FCCP (final 2 μM) were added directly to cell suspensions. The cuvette was thoroughly washed between determinations. The consumption rate was determined following addition of ethanol or FCCP as the maximum stable rate of decline in oxygen consumption. Negligible oxygen consumption was observed in the absence of ethanol.

**Assessment of mitochondrial membrane potential**. 1.0 $OD_{600}$ unit of cells was harvested, washed once in fresh culture media, and stained in the dark using DiOC6 (final concentration 1 μg/mL) for 5 min at 30 °C with constant agitation. Cells were washed three times in fresh media before being analysed on an Accuri C6 flow cytometer (BD) using BD Accuri C6 software (Ver. 1.0.264.21). As described in Supplementary Fig. 12, events were gated by removing non-cell debris by forward and side scatter data. Following this, the proportion of highly fluorescent newly emerged cells was identified by gating for the indicated region in the FL1-A channel. The proportion of cells falling within this region are shown in Fig. 5b. The logicle transformation was used for forward and side scatter data. FCCP-treated cells were used as a negative control by incubating cells for 10 min in the presence of 10 μM FCCP with agitation immediately before staining.

**Extraction of total RNA from cells**. Total RNA was extracted from cells using a modified TRIzol protocol. Briefly, 1 mL of TRIzol reagent (Life Technologies) was added to 50 $OD_{600}$ units of pelleted cells and mixed by pipetting. Zirconia beads (Yasui Kikai YZB05, diameter 0.5 mm) were then added and samples subjected to lysis in a multi-beads shocker (Yasui Kikai) at 0 °C, with four 30 sec cycles of beating at 2500 rpm/30 sec recovery between beating cycles to prevent sample heating. 200 μL of chloroform was then added, samples were subjected to centrifugation (15 min, 12,000 g, 4 °C) and the upper aqueous phase was transferred to a fresh tube. Following the addition of 1 μL of 20 mg/mL glycogen, 500 μL of isopropanol was added to precipitate RNA and allowed to incubate on ice for 30 min. Samples were then centrifuged (10 min, 12,000 g, 4 °C), the pellet washed in 75% ethanol, and following a final centrifugation ethanol was removed and samples allowed to air dry for 10 min. Pellets were resuspended in 30 μL of ultrapure $H_2O$. RNA integrity was confirmed by agarose gel electrophoresis and by spectrometry ($A_{260}/A_{280}$ ratio).

**Electrophoresis of total RNA samples**. Before subjecting samples to electrophoresis, total RNA yield was estimated by spectrometry and RNA concentrations were made equivalent by the addition of ultrapure $H_2O$ where appropriate. Due to the fragility of the assessed tRNA modification, samples must not be heated and must be run in acidic conditions. Samples were therefore prepared for electrophoresis by the addition of one volume of sample buffer (0.1 M sodium acetate, pH 5.0, 8 M urea, 0.05% bromophenol blue, 0.05% xylene cyanol) without heating before separation by acid-urea PAGE. The gel is a 6.5% acrylamide gel containing 8 M urea that is adjusted to pH 5.0 by the addition of 0.1 M sodium acetate (made to pH 5.0 with acetic acid). Polymerisation is achieved by the addition of 0.04% ammonium persulphate and 0.1% tetramethylethylenediamine. Due to the small band shift upon tRNA modification, the gel is run in a ~40 cm sequencing gel format with continuous reticulation of fresh sodium acetate buffer (1 M, pH 5.0) to maximise resolution. Samples are loaded using a shark's tooth comb, with spaces left between samples filled with loading buffer. The gel is set up and run at 4 °C, and is prewarmed by running at 500 V constant voltage for 30 min. Samples are electrophoresed at 400 V until the xylene cyanol band runs off the gel (20–22 h).

**Northern blotting of RNA to determine tRNA modification**. Separated total RNA samples were transferred to a nylon membrane using a BioRad Trans-Blot SD Semi-Dry Transfer Cell in 1X TAE buffer (2 M Tris, 1 M acetic acid, 50 mM EDTA) at room temperature (30 min, 150 mA). Following transfer, RNA was crosslinked using a Funa UV-linker (1.2 mJ/cm², Funakoshi). Pre-hybridisation was then carried out using PerfectHyb Plus hybridisation buffer, with membranes incubated in hybridisation tubes at 37 °C and 15 rpm for 30 min without blocking. A mitochondrial initiator tRNA-specific oligonucleotide, labelled at the 5' end according to the manufacturer's protocol using [γ-³²P] ATP (Perkin Elmer) and T4 polynucleotide kinase (Takara), was added directly to the hybridisation solution. Hybridisation was carried out as for pre-hybridisation for 3 h. The membrane was then washed once in low-stringency wash solution (0.3 M sodium chloride, 30 mM sodium citrate, pH 7.0, 0.1% SDS, in ultrapure water) for 5 min at room temperature, and then twice in high-stringency wash solution (0.75 M sodium chloride, 7.5 mM sodium citrate, pH 7.0, 0.1% SDS, in ultrapure water) and exposed to a phosphor imagine plate (GE healthcare) for 24 h before imaging on a FLA-7000 laser scanner (Fujifilm).

**Estimation of mitochondrial translation efficiency**. Mitochondrial translation was assessed by the established method[55] with some modifications. Briefly, 0.75 $OD_{600}$ units of cells were removed from culture at relevant time points, to which 30 μg/mL of cycloheximide (freshly prepared in DMSO) was added. After a 3 min incubation at room temperature, a solution of all proteinogenic amino acids but serine, methionine and cysteine was added (final concentration 64 μg/mL). A 10 mCi/mL solution of [³⁵S]-methionine labelling solution was then immediately added at a dilution of 1/125 and cells were left to incubate for 30 min at 30 °C. Following this, chloramphenicol (final concentration 0.5 mg/mL) and unlabelled methionine (final concentration 4 mM) were added to stop labelling. Samples were frozen and then subjected to alkaline lysis, TCA-assisted protein precipitation, a single acetone wash and subsequent resuspension in a standard SDS-PAGE sample buffer. As boiling results in the aggregation of mitochondrial proteins, samples were agitated for 30 min at room temperature to resuspend proteins. Samples were subsequently separated by standard SDS-PAGE methods on a 15% gel. Following Coomassie staining, the gel was dried and exposed to a phosphor imaging plate (GE healthcare) for several days before imaging on a FLA-7000 laser scanner (Fujifilm).

**Determination of cell size**. Cells sampled at indicated time points were briefly sonicated to separate associated cells before analysis of cell size using a Sysmex CDA-1000 series particle counter (Sysmex Corporation, Kobe, Japan) according to the manufacturer's instructions.

**Determination of cell viability**. Appropriately diluted cells were mixed well with 0.1% w/v methylene blue at a 1:1 ratio and immediately visualised by light microscopy. The proportion of living (not coloured) to dead (coloured) cells was determined using a haemocytometer. At least 300 cells were counted per strain for each replicate. Data are from two independent experiments.

**Determination of cell petite frequency**. Petite frequencies of strains were determined as described in Ogur et al.[56]. Briefly, strains were grown as indicated and removed from culture. Cell concentration was determined by haemocytometer and approximately 100 cells were spread onto YPD plates. After 4 days of incubation at 30 °C, a solution of 67 mM potassium phosphate buffer (pH 7.0) and 2% w/v agar was prepared, autoclaved and allowed to cool to approximately 55 degrees before 0.1% w/v Triphenyltetrazolium chloride (TTC, Sigma) was added and completely dissolved. Plates were then removed from the incubator and the molten TTC solution was gently pipetted over colonies to fill the plate. The proportion of normal (pink) to respiratory deficient (white) colonies was counted after 3 h incubation at room temperature.

**Estimation of culture parameters**. Fermenter and compact rocking incubator data were analysed using the R programming language (Ver 3.5.2)[57]. Culture parameters were estimated using the Grofit package (Ver. 1.1.1-1) on raw data[58]. For presentation of plots, a smoothing function based on loess fitting of raw data within the ggplot2 package (Ver. 3.2.1) was used to reduce measurement noise from compact rocking incubator data. Where culture parameters are represented as boxplots, data from at least three independent growth determinations (shown individually as points) were used.

**Statistical analyses**. All statistical analyses were performed using the R programming language (Ver 3.5.2)[57]. For comparisons between conditions, Welch's t-test was used to account for differences in variance between data sets. For the dose–response curve (Fig. 4c), a logistic curve fitting algorithm was applied to mean data from three independent determinations using the drc package (Ver. 3.0-1)[59]. Throughout the paper, averages represent mean values, and error bars represent one standard deviation from the mean. Significance was assumed where $p \leq 0.05$. No data were excluded for the purposes of statistical analyses. Boxplots represent the median (middle bar), 25th and 75th percentiles (upper and

lower limits of the box) and 1.5 × interquartile range (whiskers). Specific statistical parameters are indicated in figure legends where relevant.

**Reporting summary**. Further information on research design is available in the Nature Research Reporting Summary linked to this article.

## Data availability

Raw automated growth data and flow cytometry data are available from the corresponding author upon reasonable request. Source data are provided with this paper.

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

## Acknowledgements

We thank Trever Lithgow for the gift of the D273-10B strain, Yukihide Tomari, Katsuhiko Shirahige and Traude Beilharz for experimental support and Yasuhiro Araki, Peter Boag and the members of the Ohsumi lab for helpful discussions. This work was supported by JSPS KAKENHI Grant Numbers JP23000015 and JP16H06375 (to Y.O.) and Australian Research Council Discovery Project Grant DP130100818 (to M.P.). A.I.M. received support from a Prime Minister's Australia Asia Award.

## Author contributions

A.I.M., Y.O. and M.P. designed experiments. A.I.M. performed experiments. A.I.M. and Y.O. analysed data and wrote the manuscript.

## Competing interests

The authors declare no competing interests.
