## [Peer Review File · Nature Communications]

Reviewers' comments:

Reviewer #1 (Remarks to the Author):

The authors demonstrate the physiological significance of yeast autophagy during transition from fermentative to respiratory growth. They firstly showed that inhibition of autophagy caused a delay in resuming growth after the transition. They identified serine as a key compound for restart of the growth and for induction of mitochondrial respiratory activity. Formation of mitochondrial N-formyl-methionine-tRNA was found to be the critical step for which serine was supplemented through autophagy.

Several comments were described below for improvement of the manuscript.

1. The authors group already published several studies regarding the role of autophagy in the fermentative-to-respiratory transition. In the discussion part, such previous studies should be described to show the originality of this work, and to reinforce or deepen the arguments.
2. No statistical information for the growth monitoring is described in the manuscript. Are the growth lines plots of average values from several experiments, or the representative data from single cultures?
3. (Fig. 6b) The time periods of the growth delay with *mis1delta* and *fmt1delta* should be clearly described for both serine-supplemented and non-supplemented cultures. And importantly, the alleviation of the delay is detectable even in the two strains defective in the THF-mediated mitochondrial initiation tRNA synthesis. Do the authors have some notion to reconcile the data?

Reviewer #2 (Remarks to the Author):

In the current manuscript, the authors further explore the important question of the physiological functions of autophagy. They specifically show that prototrophic budding yeast requires macroautophagy to efficiently adapt to respiratory growth upon transitioning from amino acid rich synthetic medium containing glucose to synthetic medium without amino acids in the presence of non-fermentable carbon sources. Delayed adaptation correlates with reduced respiratory activity, reduced induction of fMet-tRNA levels and mitochondrial translation. The authors make the intriguing observation that mitochondria-associated phenotypes in *atg2* mutants are rescued specifically upon serine supplementation, but not with glycine or glutamine, and correlate the mitochondrial phenotypes with changes in one-carbon metabolism.

However, whether differences in one-carbon metabolism are causative for the impaired adaptation of *atg2* mutants to respiratory growth is not convincingly demonstrated yet. Specifically, the authors show that glycine addition does not rescue respiratory growth of *atg2* mutant cells although glycine can promote one-carbon metabolism. This observation raises the distinct possibility of alternative causes for the observed adaptive defects. Thus, in my point of view the manuscript requires substantial additional experimentation.

Conceptual criticism:

- (1) The authors should include a *shm1* mutant as comparison in figure 6, as this deletion should be more closely related to the proposed *atg2* defect. Moreover, glycine can be used in one-carbon metabolism by conversion to serine by *Shm2* in the cytosol or by the glycine decarboxylase complex (GDC) to form CH₂-THF within mitochondria (which the authors did not include in Figure 6a). However, as shown in Figure 4d, glycine supplementation did not alleviate the prolonged lag phase of *atg2* mutants. This finding is difficult to reconcile with the model proposed by the authors that reduced one-carbon metabolism in *atg2* mutants is limiting for respiratory adaptation. Thus, to critically test their model, a key experiment is to examine whether glycine supplementation rescues one-carbon

metabolism, fMet-tRNA synthesis, and mitochondrial translation and respiration in atg2 mutants, but may not affect the lag phase phenotype.

In line with this notion, does the rescue of cell growth of atg2 mutants upon serine supplementation depend on one carbon metabolism? How do atg2 shm1, atg2 fmt1 and atg2 mis1 double mutants behave? Does their growth still improve upon serine supplementation?

Figures 6c and d lack critical controls and should include the mutants shm1, fmt1 and mis1 for comparison. In Figure 6d, the loading of the lanes -ser and +ser at timepoint B is significantly lower than in all other lanes and therefore it is difficult to judge whether +ser actually improves mitochondrial translation in atg2 mutants compared with -ser. The data need to be replaced by a more equally loaded example and, importantly, need to be quantified.

(2) Translation of mitochondrial encoded proteins is coupled to cytosolic translation (Couvillion et al, Nature 2016). This raises the possibility that serine is limiting for cytosolic translation and as a downstream consequence for mitochondrial translation. To test this possibility, the authors should compare cytosolic with mitochondrial translation in wt and atg2 cells \pm serine supplementation. Given the model of the authors, serine supplementation should mainly affect mitochondrial translation and not cytosolic translation.

(3) In a previous paper, the Ohsumi lab has shown that iron mobilization by autophagy in yeast cells grown in SD medium is required for adaptation to respiratory growth (Adachi et al. JCB, 2017). Does iron supplementation rescue the growth defect of autophagy mutants in the current experimental setup?

(4) So far, the presented data indicate that cells required autophagy to adapt to respiratory growth upon low amino acid availability (shifting cells from glucose media + casamino acids to ethanol media without casamino acids, but not in the presence of external amino acids/serine in ethanol media). Thus, the media conditions used shift wt and atg2 cells not only to a different carbon source but also to low amino acid conditions. This is highlighted by the observation that Atg13 is dephosphorylated when wt cells are shifted from +CAS to -CAS medium in the presence of glucose. growth on yeast extract medium containing ethanol induces Atg13 dephosphorylation (Fig. 3d), however, autophagy mutants do not show a growth delay (Fig. 4d). Here, amino acid mobilization is unlikely to be the driving force for autophagy induction. To further carefully characterize the nutrient requirements the authors should test the following conditions: Do cells preadapted to low amino acid conditions still require autophagy to transition into respiratory growth? Using precultures of glucose media without casamino acids and shifting cells to ethanol media without casamino acids would address this question. In turn, do cells require autophagy to restart cell growth when grown in ethanol media with casamino acids and then shifted to ethanol media without casamino acids?

Additional points:

(1) The authors document an impaired transition of atg2 mutant cells upon diauxic shift in figure 1a Glucose. However, this effect is not detectable in figure 1c Glucose. What is the difference between both experiments?

(2) The interpretation of autophagy flux in figure 3a is complicated by the fact that cells in glucose media are dividing and thus dilute free GFP in their vacuoles in contrast to cells in ethanol media that display a prolonged lag phase of little growth and thus may accumulate free GFP. However, the overall GFP-Atg8 turnover rate may be fairly similar between the two conditions.

(3) Figure 3b needs to include glucose conditions and needs to be quantified.

(4) In figure 5b, it is unclear how the authors define "elevated" membrane potential. What is the cutoff for "normal" or "low" membrane potential? This needs to be clearly defined in the figure legend and/or materials and method section.

(5) It seems biased to call the effect of serine supplementation on the growth of fmt1 mutants "marginally" in figure 6b.

Reviewer #3 (Remarks to the Author):

In the manuscript entitled "Autophagy facilitates adaptation to respiratory growth by recycling serine for one-carbon metabolism" May et al present a very interesting history showing the need of non-selective autophagy for proper diauxic shift in yeast. Comparing different atg mutants and using different nutrient supplementation they found that the addition of serine in the culture media is sufficient to abbreviate $\Delta atg2$ lag phase on ethanol. They observed a deficit in OXPHOS rates in $\Delta atg2$ that is also rescued by serine. Finally, they connect this respiratory deficit in autophagy mutants to one-carbon metabolism and the supply of tetrahydrofolate necessary for proper mitochondrial translation initiation through tRNA-met formylation by Fmt1p. The authors did not discuss properly that in yeast mitochondria translation initiation can occur in the absence of Fmt1 (Li et al., 2000 J Bacteriol. 182: 2886-2892.) which is only abolished if $\Delta fmt1$ deletion is combined with $\Delta rsm28$ (Williams et al., 2007 Genetics. 175: 1117-1126.) or in the double mutants $\Delta fmt1 \Delta msc6$ (Franco et al., 2019 FEBS J 286:1407-1419) $\Delta aep3 \Delta fmt1$ (Lee et al. 2009 J Biol Chem. 284: 34116-34125) Perhaps due to the presence of too many curves in Fig 6B the understanding of the effect serine addition over $\Delta fmt1$ mutants is not very well evaluated. It would be nice to have this figure better presented. Does extra serine abbreviate the lag phase of $\Delta fmt1$, but not $\Delta mis1$? In Fig 6C the formylation of tRNA-met is increased in $\Delta atg2$ mutants in the presence of serine, but the same should be tested in the $\Delta fmt1$ mutant, as well as this mutant should be tested of newly synthesized mitochondrial products (Fig 6D). These controls are necessary to support the authors hypothesis over the use of serine in the formation of tRNA-met-formyl and not in other aspect of mitochondrial translation metabolism. If this hypothesis is sustained by the suggested controls then another conclusion of this work is that the mitochondrial formylation of tRNA-met is required for optimal diauxic shift.

Minor - Fig 1A and 4A please include the legends bars black for wt and blue for atg2
Please substitute the term "mitochondrial function" along the text to oxidative phosphorylation or OXPHOS ...

Response to reviewers: revised submission of May et al manuscript (*Autophagy facilitates adaptation to respiratory growth by recycling serine for one-carbon metabolism*, NCOMMS-19-39692-T).

We thank the reviewers for their time in assessing our work and are grateful for the constructive feedback they have kindly provided. We have conducted a series of additional experiments that we are confident have addressed the reviewers' concerns, and we feel that these amendments have greatly improved the quality of our manuscript.

Below, we provide point-by-point responses to the reviewers' comments. We have provided two versions of the revised manuscript. The 'markup' document includes all the changes to the text made in this round of revision, with changes made in direct response to reviewer comments indicated in red and small corrections of typographical errors represented in blue. The 'original' version provides the revised text without any indications of changes. Thank you again for your consideration of our manuscript.

Reviewer #1 (Remarks to the Author):

The authors demonstrate the physiological significance of yeast autophagy during transition from fermentative to respiratory growth. They firstly showed that inhibition of autophagy caused a delay in resuming growth after the transition. They identified serine as a key compound for restart of the growth and for induction of mitochondrial respiratory activity. Formation of mitochondrial N-formyl-methionine-tRNA was found to be the critical step for which serine was supplemented through autophagy.

Several comments were described below for improvement of the manuscript.

1. The authors group already published several studies regarding the role of autophagy in the fermentative-to-respiratory transition. In the discussion part, such previous studies should be described to show the originality of this work, and to reinforce or deepen the arguments.

Our group is interested in characterising the various biological roles of autophagy. We have therefore used a range of growth conditions to uncover new physiological phenotypes in autophagy mutant cells. We have previously examined the link between autophagy and respiration using two conditions: the diauxic shift and carbon starvation. In these studies, we reported the induction of autophagy during the diauxic shift, hinting at a relationship between adaptation to respiration and autophagy, and have also showed that respiratory activity is required for the activation of autophagy during carbon starvation, likely due to the provision of ATP by oxidative phosphorylation. However, neither of these conditions provides a simple experimental regime allowing the examination of the effect of fermentative to respiratory transition. In order to limit the number of variables and conduct a more direct analysis, we shifted cells to minimal media supplemented with a respiratory carbon source. This approach has allowed us to clearly demonstrate that the conversion to respiratory metabolism alone is sufficient for autophagy induction. We also show the specific metabolic requirement for serine during the transition to respiration. The identification of one-carbon metabolism as the pathway requiring this amino acid is the first time that the use of an autophagy-derived metabolite by a specific pathway has been identified. The paper therefore stands alone as an original and important contribution to the field. We have updated the discussion to include references to our group's previous works (from line 292).

2. No statistical information for the growth monitoring is described in the manuscript. Are the

growth lines plots of average values from several experiments, or the representative data from single cultures?

We thank the reviewer for pointing out this omission on our part. The data shown in the main figures of the paper are representative individual growth curves (i.e. from single cultures, with cultures plotted on the same graph always collected in the same experiment). As can be seen in Fig. 2c, there is some variability in the onset of growth between experiments. Averaging of growth curves therefore eliminates much of the detail evident in individual growth curves. For this reason, we determined that individual growth curves give the most accurate portrayal of growth. On the other hand, growth quantification was performed using data from all experiments, allowing a robust statistical analysis of the delayed onset of autophagy mutant growth described in this paper (e.g. Fig. 2c). We have updated the relevant figure legends to clarify this point (See lines 523, 534, 556, 570, 606, 613 and 623) and have indicated the statistical information where relevant in new figure legends added in this version of the manuscript.

3. (Fig. 6b) The time periods of the growth delay with mis1delta and fmt1delta should be clearly described for both serine-supplemented and non-supplemented cultures. And importantly, the alleviation of the delay is detectable even in the two strains defective in the THF-mediated mitochondrial initiation tRNA synthesis. Do the authors have some notion to reconcile the data?

We have replaced Fig. 6b with a clearer summary of the relevant growth phenotypes of WT, *atg2Δ*, *fmt1Δ* and *mis1Δ* strains. In addition, following advice from the other reviewers we have also added data for the double mutant *atg2Δ fmt1Δ* and *atg2Δ mis1Δ* strains.

Regarding the alleviation of growth onset in *mis1Δ* and *fmt1Δ* strains, we see a partial recovery of growth delay in all strains, including WT cells, under the conditions used in this study. This likely reflects that serine is utilised by other pathways in addition to one-carbon metabolism under these conditions. As multiple pathways compete for limited serine within cells, we expect that the localisation of one-carbon metabolism in the mitochondrion renders translation initiation particularly vulnerable to low concentrations of the amino acid. However, the recovery of *atg2Δ* growth is particularly marked, clearly suggesting that the flow of serine into mitochondrial one-carbon metabolism is critical for the restoration of *atg2Δ* respiratory growth. We have included discussions of this point in the main text at lines 221 and 331.

Reviewer #2 (Remarks to the Author):

*In the current manuscript, the authors further explore the important question of the physiological functions of autophagy. They specifically show that prototrophic budding yeast requires macroautophagy to efficiently adapt to respiratory growth upon transitioning from amino acid rich synthetic medium containing glucose to synthetic medium without amino acids in the presence of non-fermentable carbon sources. Delayed adaptation correlates with reduced respiratory activity, reduced induction of fMet-tRNA levels and mitochondrial translation. The authors make the intriguing observation that mitochondria-associated phenotypes in *atg2* mutants are rescued specifically upon serine supplementation, but not with glycine or glutamine, and correlate the mitochondrial phenotypes with changes in one-carbon metabolism.*

*However, whether differences in one-carbon metabolism are causative for the impaired adaptation of *atg2* mutants to respiratory growth is not convincingly demonstrated yet. Specifically, the authors show that glycine addition does not rescue respiratory growth of *atg2**

mutant cells although glycine can promote one-carbon metabolism. This observation raises the distinct possibility of alternative causes for the observed adaptive defects. Thus, in my point of view the manuscript requires substantial additional experimentation.

Conceptual criticism:

(1) The authors should include a *shm1* mutant as comparison in figure 6, as this deletion should be more closely related to the proposed *atg2* defect. Moreover, glycine can be used in one-carbon metabolism by conversion to serine by *Shm2* in the cytosol or by the glycine decarboxylase complex (GDC) to form CH₂-THF within mitochondria (which the authors did not include in Figure 6a). However, as shown in Figure 4d, glycine supplementation did not alleviate the prolonged lag phase of *atg2* mutants. This finding is difficult to reconcile with the model proposed by the authors that reduced one-carbon metabolism in *atg2* mutants is limiting for respiratory adaptation. Thus, to critically test their model, a key experiment is to examine whether glycine supplementation rescues one-carbon metabolism, *fMet*-tRNA synthesis, and mitochondrial translation and respiration in *atg2* mutants, but may not affect the lag phase phenotype.

We thank the reviewer for the suggestion that we further clarify the link between autophagy and one-carbon metabolism. The reviewer raises two points, the first concerning *Shm1* and the second the role of glycine in one-carbon metabolism. We deal with the comment regarding *Shm1* first.

We constructed an *shm1*Δ strain to test whether the deletion of this locus results in a similar growth phenotype to *atg2*Δ cells. As shown in the new Fig. S9a, while the delay in onset of respiratory growth was comparable to WT cells, overall growth on respiratory media was ‘flattened,’ with a characteristic slowing of growth rate (growth on glucose was normal). As the reviewer will be aware, *Shm1* supplies one-carbon units within the mitochondrion through a hydroxymethyltransferase reaction that uses serine as a reactant and produces glycine as a product. As shown in Fig. 6a, one-carbon metabolism is characterised by the linkage of cytosolic and mitochondrial branches, and there also exists a cytosolic serine hydroxymethyltransferase that has been reported to supply up to 95% of one-carbon units in cells under fermentative conditions (McNeil et al. (1994) *JBC*, Kastanos et al. (1997) *Biochemistry*). One-carbon metabolic units may therefore flow from the cytosol to mitochondria under the conditions assessed in this report, especially in the absence of *Shm1*. While an early report (Zelikson and Luzzati (1976) *Eur. J. Biochem*) showed that a temperature-sensitive *shm1* mutant strain was unable to grow by respiration, subsequent genome-wide mutant screens have not reported any defects in respiratory growth phenotypes for *shm1*Δ strains, suggesting that mitochondrial one-carbon metabolism is not seriously affected by *SHM1* deletion (this point is discussed in terms of formylation below in our response to the reviewer’s comment about controls for the formylation data). The *shm1*Δ result is in contrast with the growth of *mis1*Δ and *fmt1*Δ strains, which are critical for the reactions of mitochondrial one-carbon metabolism. We therefore conclude that the cyclical, reversible nature of one-carbon metabolism accounts for the discrepancy between *shm1*Δ and *atg2*Δ (and, equally, *mis1*Δ and *fmt1*Δ) growth phenotypes. In addition to the new supplementary Fig. S9, we have added a comment about the *SHM1* phenotype at line 226 and have also updated Fig. 6a to better illustrate the possibility of one-carbon metabolic flow from the cytoplasm to mitochondria.

With regard to the potential use of glycine as a one-carbon precursor, we assessed the formylation of WT and *atg2*Δ cells in the presence of glycine (otherwise under the same conditions as Fig. 6c), finding no recovery in formylation upon supplementation of glycine (Fig. S11c). Glycine can be used as a one-carbon metabolic precursor through the activity of the serine

hydroxymethyltransferases (discussed above) or, as the reviewer points out, through the glycine cleavage pathway. However, the formylation data together with the non-specific inhibition of growth shown in the original Fig. 4d suggests to us that under the conditions used in this study, glycine may in fact inhibit one-carbon metabolism in general through product inhibition of the Shm enzymes. We note that the conditions employed in this study differ from previous studies, which may account for the apparent discrepancy regarding the ability of glycine to act as a one-carbon metabolism substrate. We have added a new paragraph that discusses the role of glycine and this result at line 252 and have updated Fig. 6a to better illustrate the potential contribution of the glycine cleavage complex to mitochondrial one-carbon metabolism.

In line with this notion, does the rescue of cell growth of atg2 mutants upon serine supplementation depend on one carbon metabolism? How do atg2 shm1, atg2 fmt1 and atg2 mis1 double mutants behave? Does their growth still improve upon serine supplementation?

We tested the growth of *atg2Δ shm1Δ*, *atg2Δ fmt1Δ* and *atg2Δ mis1Δ* strains following the reviewer's suggestion. These data are shown in the new Fig. 6b, which we have updated to enhance the readability of the figure, and Fig. S9b.

In line with the limited effect of *SHM1* deletion on lag phase duration, the *atg2Δ shm1Δ* strain exhibited similar delays in respiratory growth onset to *atg2Δ* cells (including the characteristic recovery of growth upon serine supplementation). This strain also showed the slow growth rate observed for the single *shm1Δ* mutant. As discussed in our response to point 1, we interpret these data as indicating that the reactions of cytosolic one-carbon metabolism are able to utilise serine and deliver one-carbon metabolites to the mitochondrion.

Meanwhile, *atg2Δ fmt1Δ* and *atg2Δ mis1Δ* strains showed a moderate prolongation in respiratory growth lag duration in comparison to single *atg2Δ* mutant cells. As observed for *fmt1Δ* and *mis1Δ* single deletion strains (and indeed WT cells), serine supplementation was able to recover the growth of double deleted strains in combination with *atg2Δ*. However, the serine-mediated recovery of growth was reduced in comparison to *atg2Δ* cells. Notably, the double mutant strains were recovered only to the lag durations of their respective one-carbon single deletions strains in the absence of serine (i.e., *atg2Δ fmt1Δ* (+ serine) → *fmt1Δ* (- serine) and *atg2Δ mis1Δ* (+ serine) → *mis1Δ* (- serine)), suggesting that the additional lag observed in these strains and the effect of serine supplementation is due to some other, one-carbon metabolism independent pathway.

It is expected that in *atg2Δ* cells exposed to amino acid starvation conditions, the intracellular amino acid pool – including serine – is generally depleted; indeed, this is the very reason that we are able to interrogate the role of serine in adaptation to respiratory growth. The depletion of amino acids is certain to have an effect on various pathways in the cell, including translation (discussed below), and the data may suggest that the charging of Ser-tRNA by free serine may also have an effect on growth, therefore indicating that serine is required for both initiation and elongation of translation. However, such a scenario is also very likely to further limit the availability of free serine to one-carbon metabolism. To definitively address the relative contributions of various pathways that compete for limited serine, additional data assessing the transport of amino acids into the mitochondrion and the concentrations of amino acids within intracellular compartments would be necessary. However, such questions are beyond the scope of the current report, which provides clear evidence that serine depletion results in an inability to initiate mitochondrial translation during the transition to respiratory growth. We have added a discussion of these growth phenotypes at line 220, as well as a comment emphasising that the

non-specific recovery of growth may be due to the utilisation of serine by other pathways at line 331.

*Figures 6c and d lack critical controls and should include the mutants *shm1*, *fnt1* and *mis1* for comparison. In Figure 6d, the loading of the lanes -ser and +ser at timepoint B is significantly lower than in all other lanes and therefore it is difficult to judge whether +ser actually improves mitochondrial translation in *atg2* mutants compared with -ser. The data need to be replaced by a more equally loaded example and, importantly, need to be quantified.*

We have replaced Fig. 6d with data from a new experiment. In this experiment, we increased the number of assessed time points and have ensured that the loading is consistent across the gel. We have also included a panel indicating the intensity of bands observed on the gel. From these data, we draw the following conclusions: (1) the trend of delayed *atg2*Δ mitochondrial protein expression was reproduced in this experiment; (2) as observed in the original Fig. 6b, serine supplementation enhances mitochondrial expression, suggesting that it is limiting for mitochondrial translation under these conditions; and (3) *atg2*Δ mitochondrial protein expression is enhanced in the presence of serine, and is brought to a level comparable to WT cells at time point C, when *atg2*Δ and WT growth rates are comparable. These data therefore confirm that serine supplementation supports mitochondrial translation in *atg2*Δ cells.

We have also included two new supplementary figures demonstrating the formylation of initiator tRNA in *shm1*Δ, *fnt1*Δ and *mis1*Δ strains (Fig. S11b and c). As expected, these results indicate that formylation is completely blocked in *fnt1*Δ and *mis1*Δ strains, with uncharged and charged but not formylated tRNAs appearing in these strains. In contrast, formylation in the *shm1*Δ strain appears to be comparable to WT cells at early time points, and slightly lower at time point C (following the onset of growth). These phenotypes are consistent with the growth phenotypes of the strains described. We outline these findings at lines 242 and 246.

With regard to the mitochondrial expression controls suggested by the reviewer, we have included new data in Fig S11d. Unfortunately, this blot (which shows extensive cracking that arose during drying) is the only data we were able to collect before the radioactive research facility of our university was completely shut down due to ongoing Covid-19-related lockdowns in Japan. The recovery of growth by serine was also uncharacteristically weak in this experiment. We regret that we were not able to collect higher quality data for the reviewer. In general, this result shows that translation efficiency reflects the formylation state of mitochondrial initiator tRNA: whereas WT and *shm1*Δ cells (both characterised by normal formylation) show robust protein expression, *atg2*Δ, *fnt1*Δ and *mis1*Δ strains all showed reductions in comparison to WT cells. The supplementation of serine was able to restore *atg2*Δ expression, but had little impact on *fnt1*Δ and *mis1*Δ expression at all time points assessed. These results provide strong evidence that mitochondrial initiator tRNA formylation, mitochondrial translation and the timely onset of respiratory growth are tightly linked. We describe these results at line 272.

*(2) Translation of mitochondrial encoded proteins is coupled to cytosolic translation (Couvillion et al, Nature 2016). This raises the possibility that serine is limiting for cytosolic translation and as a downstream consequence for mitochondrial translation. To test this possibility, the authors should compare cytosolic with mitochondrial translation in wt and *atg2* cells ± serine supplementation. Given the model of the authors, serine supplementation should mainly affect mitochondrial translation and not cytosolic translation.*

The shutdown of radioactive research facilities has unfortunately prevented us from collecting

additional detailed quantitative data that precisely addresses this comment. However, as an alternative approach we directly assessed the protein levels of two mitochondrial proteins, Cox2 (mitochondrially encoded) and Atp2 (encoded on the genome), by western blotting, as shown below in Fig. R1.

Fig. R1. Western blotting of mitochondrial proteins Cox2 (mitochondrially-encoded) and Atp2 (nuclear-encoded) using α -Cox2 and α -Atp2 antibodies. Samples were removed from culture at the indicated time points (top panel) and subjected to western blotting (lower panels). Actin is provided as a loading control.

These data show that in contrast to the clear delay in mitochondrial protein expression, Atp2 levels are comparable between WT and *atg2Δ* cells (note that the greater resistance of cells to lysis during respiratory growth, as reflected in actin protein levels, masks the increase in Atp2 protein levels). This result is consistent with the very uniform whole cell protein levels observed by Coomassie staining of 35S labelling samples, as shown in Figs. 6d and S11d.

In addition to these data, we strongly favour the proposed model, whereby serine feeds into mitochondrial translation, for two key reasons. First, as our growth regime employs respiratory growth media lacking amino acids, it is very unlikely that serine alone is limiting for translation. The intracellular concentration of serine is intermediate among amino acids, with other amino acids such as tryptophan, methionine, cysteine and phenylalanine characterised by much smaller intracellular pools than serine (see Kitamoto et al. (1988) *Journal of Bacteriology* and Müllerer et al. (2016) *Cell*). Following this logic, these amino acids should also be very likely to be limiting for translation, which we did not find to be the case in our initial screen for amino acids affecting *atg2Δ* growth. In addition, if one-carbon metabolism is indeed unrelated to the recovery of autophagy mutant growth, one might expect that the activity of the serine hydroxymethyltransferases would supply serine upon supplementation of glycine, which we did not observe (Fig. 4d), likely for the reasons discussed in our response to point 1.

Second, the abrupt withdrawal of a fermentative carbon source is associated with a depletion of intracellular ATP, as previously shown by our own group (Adachi et al. (2017) *JBC*) and more

recently (Weber et al. (2020) *PNAS*). The ability to commence and maintain cytosolic translation in such a scenario depends on the cell's ability to generate ATP through oxidative phosphorylation. In addition, the anaplerotic reactions of the TCA cycle generate amino acids, which are required for translation. This means that while a direct mechanism for the mitochondrial regulation of cytoplasmic translation may not exist, the absence of mitochondrial function observed in our study results in an energetic and biochemical situation that may severely inhibit bulk protein synthesis. This interpretation of the data is supported by the normal growth rate observed for *atg2Δ* cells once growth actually commences. For these reasons, we don't think that serine alone is specifically limiting for translation. We have added a paragraph discussing the link between cytoplasmic and mitochondrial translation mentioned by the reviewer and cites the study that they kindly brought to our attention (see line 337).

(3) In a previous paper, the Ohsumi lab has shown that iron mobilization by autophagy in yeast cells grown in SD medium is required for adaptation to respiratory growth (Adachi et al. JCB, 2017). Does iron supplementation rescue the growth defect of autophagy mutants in the current experimental setup?

Horie et al. (*JCB*, 2017) assessed the growth of autophagy mutant cells after the diauxic shift, finding that the supplementation of iron allowed autophagy mutant cells to achieve higher growth yields. In line with the reviewer's suggestion, we tested for the ability of iron to rescue the delayed onset of autophagy mutant growth and found that the supplementation of iron at the same concentrations as described in Horie et al. (op. cit.) had no effect on the delayed onset of growth (Fig. S8). We describe this result at line 168.

(4) So far, the presented data indicate that cells required autophagy to adapt to respiratory growth upon low amino acid availability (shifting cells from glucose media + casamino acids to ethanol media without casamino acids, but not in the presence of external amino acids/serine in ethanol media). Thus, the media conditions used shift wt and atg2 cells not only to a different carbon source but also to low amino acid conditions. This is highlighted by the observation that Atg13 is dephosphorylated when wt cells are shifted from +CAS to -CAS medium in the presence of glucose. growth on yeast extract medium containing ethanol induces Atg13 dephosphorylation (Fig. 3d), however, autophagy mutants do not show a growth delay (Fig. 4d). Here, amino acid mobilization is unlikely to be the driving force for autophagy induction. To further carefully characterize the nutrient requirements the authors should test the following conditions: Do cells preadapted to low amino acid conditions still require autophagy to transition into respiratory growth? Using precultures of glucose media without casamino acids and shifting cells to ethanol media without casamino acids would address this question. In turn, do cells require autophagy to restart cell growth when grown in ethanol media with casamino acids and then shifted to ethanol media without casamino acids?

The reviewer points out that two variables are implicated in the growth regime outlined in our paper: a change in carbon source and a change in amino acid availability. To properly account for both variables, we tested the growth of WT and *atg2Δ* cells under the conditions suggested by the reviewer, the results of which are shown in the new Fig. S3. The shift from SD to SE media resulted in a longer delay in the onset of autophagy mutant growth, indicating that the lag cannot be explained by the shift from casamino acid-containing media to minimal media. In contrast, only a small delay in the onset of *atg2Δ* growth was observed when cells were precultured on respiratory media. Interestingly, the delay in *atg2Δ* growth onset was reduced when cells were precultured on media containing casamino acids (SDCA or SECA), suggesting that autophagy

mutant cells precultured in the presence of casamino acids might be able to build up a pool of amino acids that are utilised during subsequent growth. These new data are described at line 76.

We also assessed the phosphorylation of Atg13 in cells transferred from SD to SE (again to exclude the contribution of amino acid downshift), as shown in the new Fig. S6b. This experiment reveals that for glucose-grown cells, dephosphorylation of Atg13 is not observed when cells are shifted to SD medium. In contrast, the SD > SE shift results in strong dephosphorylation that is comparable with rapamycin treatment (which completely inhibits TORC1 activity). These results show conclusively that the delayed onset of autophagy mutant growth reported in this study is a consequence of the shift from a fermentative to a respiratory carbon source, and that this condition results in strong autophagy induction. Phosphorylation data are described at line 130.

Additional points:

(1) The authors document an impaired transition of atg2 mutant cells upon diauxic shift in figure 1a Glucose. However, this effect is not detectable in figure 1c Glucose. What is the difference between both experiments?

We initially identified a potential respiratory growth defect in cells undergoing the diauxic shift, but subsequently found that the growth phenotypes of both WT and *atg2Δ* cells can be unreliable, especially with regard to the differences in post-diauxic growth and the final yield at the conclusion of post-diauxic growth (as pointed out by the reviewer in Figs. 1a and c). We have not determined why this is the case. Instead, we decided to shift cells to completely fresh respiratory media to directly interrogate the relationship between autophagy and respiratory growth. As shown in Fig. 2c, our revised approach is very reproducible and robust. We therefore do not consider the lack of reproducibility around the diauxic shift to be relevant to the findings of this study.

(2) The interpretation of autophagy flux in figure 3a is complicated by the fact that cells in glucose media are dividing and thus dilute free GFP in their vacuoles in contrast to cells in ethanol media that display a prolonged lag phase of little growth and thus may accumulate free GFP. However, the overall GFP-Atg8 turnover rate may be fairly similar between the two conditions.

Cleavage assays such as that presented in Fig. 3a are complicated by cellular division, which as the reviewer notes results in the ‘dilution’ of both the full-length chimeric protein and the cleavage product. In Fig. 3a the accumulation of GFP-Atg8 cleavage products and Ape1 processing products in ethanol-grown cells is apparent from 3.5 h, a point at which glucose-based growth has only just commenced and dilution is unlikely to be a major factor. In order to further clarify this point experimentally, we have also collected data assessing PAS formation in glucose-grown cells (discussed in our response to point 3 below), which indicates that autophagy is not induced during the early phase of growth on glucose (see the new Fig. S6a, which is mentioned in the text at line 120). For these reasons, we do not think that dilution of free GFP masks induction of autophagy on glucose.

(3) Figure 3b needs to include glucose conditions and needs to be quantified.

We have collected data for glucose-grown cells under otherwise identical conditions to Fig. 3b, which are now shown in the new Fig. S6a. To briefly summarise these data, PAS formation is observed in both WT and *atg2Δ* cells, which is expected because the Cvt pathway is constitutively active in WT and *atg2Δ* cells, as described in the initial submission (line 106 in the revised

version). PAS formation depends entirely upon bulk autophagy induction in *atg11Δ* cells. We discount the possibility of autophagy induction on glucose as very little PAS formation and negligible accumulation of free GFP in the vacuole are observed in these cells. In addition, the overall glucose-grown GFP-Atg8 fluorescence signal is very weak at all considered time points, indicating that the induction of Atg8 expression that accompanies autophagy induction (Kirisako et al., *JCB*, 1999) does not occur to a considerable extent in these cells. These findings, which are consistent with our Ape1 maturation and Atg13 phosphorylation data, confirm that autophagy is not induced following inoculation to glucose and are noted at line 120.

With regard to quantification, as we failed to detect any appreciable autophagic activity in glucose-grown cells, we do not think that the supplementation of quantitative data is necessary for the purposes of comparison. For these reasons, we have decided to supplement the manuscript with Fig. S6b and not quantify the data shown in this figure or Fig. 3b. We trust that the reviewer will understand our position regarding Fig. 3 based on our findings and the nature of the data.

(4) In figure 5b, it is unclear how the authors define “elevated” membrane potential. What is the cutoff for “normal” or “low” membrane potential? This needs to be clearly defined in the figure legend and/or materials and method section.

We thank the reviewer for pointing out that our description of the flow cytometry data was vague. We based our approach in the method outlined in Hughes *et al.* (Elife, 2016), whereby we initially gated at least 50,000 events to eliminate non-cell debris and then quantified the proportion of cells stained by DiOC6 that underwent a shift in fluorescence intensity. For the revised manuscript, we have decided to refine this approach. First, we transformed forward and side scatter data using a logicle transformation to remove non-cell debris, as shown in the new Fig. S12 (upper panel). We then plotted data in two dimensions (logicle-transformed forward scatter and raw DiOC6 fluorescence intensity) and gated for smaller, strongly fluorescent cells (larger cells tend to be fluorescent when analysed using DiOC6 as the dye accumulates in a cell size-dependent manner), as shown in Fig. S12 (lower panel). The percentage of cells in this subset is represented in figure 5b. We note fluorescence is observed in larger *atg2Δ* cells: larger cells accumulate more DiOC6 and thereby report a stronger signal (this signal is spread lower than in WT cells). Eliminating smaller cells with low membrane potential, which can be seen in the *atg2Δ* sample in the region beneath the gated area, has significantly reduced the background observed in the original Fig. 5b. The key finding of this experiment – a delay in mitochondrial membrane potentiation in *atg2Δ* cells that can be recovered by serine supplementation – remains unchanged. We have added the data shown above as the new Fig. S12 to make this strategy clear to readers and have updated the methods section and figure legend of Fig. 5 to guide interested readers to this new figure.

*(5) It seems biased to call the effect of serine supplementation on the growth of *fmt1* mutants “marginally” in figure 6b.*

We apologise if our wording appeared biased. We have amended the text to a more cautious interpretation of the data (from line 220).

Reviewer #3 (Remarks to the Author):

In the manuscript entitled "Autophagy facilitates adaptation to respiratory growth by recycling serine for one-carbon metabolism" May et al present a very interesting history showing the need of non-selective autophagy for proper diauxic shift in yeast. Comparing different atg mutants and using different nutrient supplementation they found that the addition of serine in the culture media

is sufficient to abbreviate *atg2* lag phase on ethanol. They observed a deficit in OXPPOS rates in *atg2* that is also rescued by serine. Finally, they connect this respiratory deficit in autophagy mutants to one-carbon metabolism and the supply of tetrahydrofolate necessary for proper mitochondrial translation initiation through tRNA-met formylation by Fmt1p.

The authors did not discuss properly that in yeast mitochondria translation initiation can occur in the absence of Fmt1 (Li et al., 2000 *J Bacteriol.* 182: 2886-2892.) which is only abolished if *fmt1* deletion is combined with *rsm28* (Williams et al., 2007 *Genetics.* 175: 1117-1126.) or in the double mutants *fmt1 msc6* (Franco et al., 2019 *FEBS J* 286:1407-1419) *aep3 fmt1* (Lee et al. 2009 *J Biol Chem.* 284: 34116-34125)

We have amended the text to better discuss the studies mentioned by the reviewer, discussing this point in further detail in the new paragraph that begins at line 204. Thank you for helping us to provide a better background to readers.

Perhaps due to the presence of too many curves in Fig 6B the understanding of the effect serine addition over *fmt1* mutants is not very well evaluated. It would be nice to have this figure better presented.

We have replaced the original figure with a simplified figure that clearly shows the lag times of the strains in question. This new figure also shows the lag periods of *atg2Δ fmt1Δ* and *atg2Δ mis1Δ* cells following the suggestion of another reviewer. We have amended our description of these data in the paragraph that begins at line 216 (specifically at line 220) and hope that these changes have clarified the data.

Does extra serine abbreviate the lag phase of *fmt1*, but not *mis1*? In Fig 6C the formylation of tRNA-met is increased in *atg2* mutants in the presence of serine, but the same should be tested in the *fmt1* mutant, as well as this mutant should be tested of newly synthesized mitochondrial products (Fig 6D).

In general, the supplementation of serine slightly reduces the lag period of all cells assessed, including WT cells, suggesting that the recovery of growth does not depend exclusively on one-carbon metabolism. There is indeed a slight serine-mediated recovery in the lag period of *fmt1Δ* cells that is less pronounced in *mis1Δ* cells. This discrepancy may be due to the broader role that *MIS1* plays in cellular one-carbon metabolism. Critically, however, the recovery of growth in both strains is dramatically reduced in comparison to *atg2Δ* cells. In line with the reviewer's suggestion, we assessed the formylation of initiator tRNA in both *fmt1Δ* and *mis1Δ* strains (shown in the new Fig. S11b-c). These data show that formylation is completely blocked in both of these strains: only unformylated and uncharged tRNAs were detected. This finding is consistent with the proposed model, whereby formylation still occurs in *atg2Δ* cells but is limited by the supply of autophagy-derived serine. Meanwhile, assessment of mitochondrial translation products (the new Fig. S11d; we were unfortunately only able to run this gel before radioactivity research facilities were closed down due to the ongoing Covid-19 pandemic and apologise for the poor presentation of these gels) showed that while not completely eliminated, mitochondrial translation is reduced in *fmt1Δ* and *mis1Δ* cells. This reflects the formylation defect of these strains and reproduces previous reports. Further, in contrast to *atg2Δ* cells, serine had no appreciable effect on mitochondrial expression in *fmt1Δ* and *mis1Δ* strains. These data lend further support to the proposed link between autophagy and one-carbon metabolism. We have added a note about the potential one-carbon metabolism independent roles for serine in the discussion at line 331.

These controls are necessary to support the authors hypothesis over the use of serine in the

formation of tRNA-met-formyl and not in other aspect of mitochondrial translation metabolism. If this hypothesis is sustained by the suggested controls then another conclusion of this work is that the mitochondrial formylation of tRNA-met is required for optimal diauxic shift.

We agree that these data have further strengthened the conclusions of this study. While we did not directly test for a link between diauxic shift (i.e. in conditions where no change in media occurred), we have added a point to the discussion to further emphasise the link between one-carbon metabolism and the adaptation to respiratory growth at line 310.

Minor - Fig 1A and 4A please include the legends bars black for wt and blue for atg2

We have added an additional set of legend bars to Fig. 1A, and have also added a legend bar to Fig. 4a. Thank you for helping us improve the clarity of our data.

Please substitute the term "mitochondrial function" along the text to oxidative phosphorylation or OXPHOS ...

We thank the reviewer for helping us to clarify our terminology and have amended the manuscript to reflect this point (see lines 174, 176 and in the figure legend of Fig. 5). While we agree that the term 'mitochondrial function' is vague, we did not directly assess oxidative phosphorylation in this study. We have therefore used the term 'mitochondrial respiration' to more accurately describe the data.

REVIEWERS' COMMENTS:

Reviewer #1 (Remarks to the Author):

The authors responded to all of my comments with additional experiments, and the manuscript is now acceptable for publication.

Reviewer #2 (Remarks to the Author):

The authors addressed all my questions.

Reviewer #3 (Remarks to the Author):

The authors answered the questions clearly and greatly improved the manuscript.